# PM-KVQ: Progressive Mixed-precision KV Cache Quantization for Long-CoT LLMs

**Tengxuan Liu**[*1,2], **Shiyao Li**[†*1,2], **Jiayi Yang**[*3], **Tianchen Zhao**[1], **Feng Zhou**[4],
**Xiaohui Song**[4], **Guohao Dai**[5,2], **Shengen Yan**[2], **Huazhong Yang**[1], **Yu Wang**[‡1]

[1]Tsinghua University, [2]Infinigence-AI, [3]Columbia University,
[4]OPPO AI Center, Beijing, China, [5]Shanghai Jiaotong University

## Abstract

Recently, significant progress has been made in developing reasoning-capable Large Language Models (LLMs) through long Chain-of-Thought (CoT) techniques. However, this long-CoT reasoning process imposes substantial memory overhead due to the large Key-Value (KV) Cache memory overhead. Post-training KV Cache quantization has emerged as a promising compression technique and has been extensively studied in short-context scenarios. However, directly applying existing methods to long-CoT LLMs causes significant performance degradation due to the following two reasons: (1) **Large cumulative error**: Existing methods fail to adequately leverage available memory, and they directly quantize the KV Cache during each decoding step, leading to large cumulative quantization error. (2) **Short-context calibration**: Due to Rotary Positional Embedding (RoPE), the use of short-context data during calibration fails to account for the distribution of less frequent channels in the Key Cache, resulting in performance loss. We propose **P**rogressive **M**ixed-Precision **KV** Cache **Q**uantization (**PM-KVQ**) for long-CoT LLMs to address the above issues in two folds: (1) To reduce cumulative error, we design a progressive quantization strategy to gradually lower the bit-width of the KV Cache in each block. Then, we propose block-wise memory allocation to assign a higher bit-width to more sensitive transformer blocks. (2) To increase the calibration length without additional overhead, we propose a new calibration strategy with positional interpolation that leverages short calibration data with positional interpolation to approximate the data distribution of long-context data. Extensive experiments on 7B–70B long-CoT LLMs show that PM-KVQ improves reasoning benchmark performance by up to 8% over SOTA baselines under the same memory budget and achieves 2.73–5.18× throughput over the original 16-bit LLMs. Our code is available at https://github.com/thu-nics/PM-KVQ.

## 1 Introduction

Recently, many pioneers have developed remarkable reasoning Large Language Models (LLMs) with long Chain-of-Thoughts (CoT) techniques, such as OpenAI-o1 (OpenAI, 2024), DeepSeek-R1 (Guo et al., 2025), QwQ (Team, 2025), and so on. To achieve better algorithmic performance, these long-CoT reasoning LLMs are trained to generate up to 128K tokens with multiple complex rationales from different perspectives (Guo et al., 2025). However, this long-CoT process demands significant memory overhead (∼10GB–100GB) to store the Key-Value (KV) Cache as the history information, which limits the practical application scenarios for such long-CoT LLMs.

To mitigate the substantial memory overhead of long-CoT LLMs, various KV Cache compression methods have been proposed (Liu et al., 2024c; Yang et al., 2024; Su et al., 2025; Xiao et al.,

---

[*]Equal contribution.

[†]Program leader.

[‡]Corresponding author: Yu Wang (yu-wang@tsinghua.edu.cn).

2023; Fu et al., 2024). Among them, Post-training KV Cache Quantization is a promising compression technique that has already been well explored in short-context scenarios (e.g., <8K tokens). QServe (Lin* et al., 2024) and MiKV (Yang et al., 2024) observe that the Key Cache has more outliers than the Value Cache, leading to higher quantization error. More importantly, the outliers in the Key Cache persist in certain channels. To this end, they propose a channel-wise equalization method to migrate the outliers from the Key tensor to the Query tensor, thereby significantly reducing the quantization error. KIVI (Liu et al., 2024c), SKVQ (Duanmu et al., 2024), and IntactKV (Liu et al., 2024b) gain insights from the data distribution of the attention map and preserve the first or most recent tokens in higher bit-width within the KV Cache to maintain the performance.

However, directly applying the above short-context-optimized methods to long-CoT LLMs results in severe performance degradation. The reasons stem from the following two aspects: (1) **Large cumulative error in long-CoT LLMs**: As a lossy compression method, directly quantizing the Key and Value tensors (Liu et al., 2024c; Lin* et al., 2024; Yang et al., 2024; Duanmu et al., 2024) introduces quantization errors at each decoding step when generating one token. As the number of generated tokens increases, the accumulated quantization error grows larger, leading to a significant performance degradation of long-CoT LLMs. (2) **Short calibration data cannot reflect long-context data distribution**: The Rotary Positional Embedding (RoPE) operator incorporates positional information into each channel of the Key Cache by rotating token embeddings using sine and cosine functions of different frequencies. For low-frequency channels after RoPE, which have a period of over 32K tokens, calibration using short sequences (e.g., 2K tokens) fails to accurately reflect the data distribution of the Key Cache, leading to more significant quantization errors.

In this paper, we propose **P**rogressive **M**ixed-Precision **KV** Cache **Q**uantization (**PM-KVQ**) to address the above two issues respectively. (1) To reduce cumulative error, we aim to fully utilize the memory budget of the target hardware through two strategies. On the one hand, we propose to quantize the KV Cache progressively. For example, to achieve extremely low-bit quantization, such as 2-bit, instead of directly quantizing KV Cache to 2-bit at each decoding step, we initially store KV Cache in 16-bit format and then gradually reduce the bit-width to 2-bit through shifting operations once the memory resource is fully occupied. On the other hand, we propose a block-wise memory allocation technique to allocate higher bit-widths for more sensitive blocks. Specifically, we formalize the bit-width allocation task as an Integer Programming problem, which can be effectively solved by existing solvers with negligible latency. (2) To increase the effective calibration length without introducing additional computational or memory overhead, we retain the use of short-context data during calibration to maintain low resource consumption. Furthermore, we propose leveraging positional interpolation (Chen et al., 2023) to embed long-context positional information into short-context data, thereby enabling a more accurate estimation of the data distribution for long sequences.

To sum up, the proposed PM-KVQ mainly contains the following contributions:

- We design progressive quantization and block-wise memory allocation techniques tailored for long-CoT scenarios to fully utilize the memory budget of the target hardware and effectively reduce cumulative quantization error.

- We propose to use short-context calibration data with positional interpolation to increase the effective length without incurring additional computational or memory overhead.

- Extensive experiments on long-CoT LLMs, ranging from 7B to 70B, show that the proposed PM-KVQ achieves up to 8% accuracy improvement over SOTA baselines on reasoning benchmarks under 4-bit/2-bit KV Cache quantization settings, while delivering a 2.73–5.18× throughput improvement over the 16-bit model.

## 2 RELATED WORK

### 2.1 LONG COT LARGE LANGUAGE MODELS

Long-CoT (Long-Chain-of-Thought) LLMs aim to enhance multi-step reasoning capabilities for complex tasks like mathematical proofs, scientific reasoning, and multi-hop QA. Models such as OpenAI-o1 (OpenAI, 2024), QwQ (Team, 2025), and DeepSeek-R1 (Guo et al., 2025) employ advanced techniques to extend CoT reasoning depth. DeepSeek, specifically, integrates iterative self-refinement and tool-augmented reasoning (e.g., code execution and symbolic solvers) to maintain

coherence across extended reasoning chains. Its architecture emphasizes hierarchical decomposition of problems and error-correction mechanisms, achieving state-of-the-art performance.

Table 1: The memory overhead of the long-CoT LLMs. The batch size is 16, and the context length is 32K.

| Model | Weights (GB) | KV Cache (GB) |
|---|---|---|
| DeepSeek-LLaMA-8B | 16 | 64 |
| DeepSeek-Qwen-32B | 64 | 128 |
| DeepSeek-LLaMA-70B | 140 | 160 |

While long-CoT can significantly improve model performance, it introduces excessively more decoding tokens (e.g., >32K tokens per request) and large GPU memory overhead. As shown in Table 1, despite employing efficient attention designs, such as Multi-Query Attention (MQA) (Shazeer, 2019), Group-Query Attention (GQA) (Ainslie et al., 2023), and Multi-head Latent Attention (MLA) (Liu et al., 2024a), the memory overhead of the KV Cache in long-CoT LLMs remains significantly large, often surpassing that of the model weights. Consequently, reducing the memory overhead of the KV Cache is important for large batch sizes and long context requirements.

## 2.2 POST-TRAINING KV CACHE QUANTIZATION

To alleviate the large memory overhead with long reasoning contexts, many efforts have been made to reduce the KV Cache size. Post-training KV Cache quantization stands as a promising technique for efficient inference. KV Cache quantization methods try to use low bit-width integers to represent the cached key and value states, instead of using high bit-width floating-point values. Existing methods typically apply asymmetric uniform quantization for KV Cache:

$$\mathbf{X}_{\text{asym}} = \left\lfloor \frac{\mathbf{X}_{\text{BF16}} - Z}{S_{\text{asym}}} \right\rceil, \tag{1}$$

$$S_{\text{asym}} = \frac{\max(\mathbf{X}_{\text{BF16}}) - Z}{2^b - 1}, \tag{2}$$

where $\mathbf{X}_{\text{BF16}}$ denotes the 16-bit brain floating point (BF16) Key or Value tensor, $\mathbf{X}_{\text{asym}}$ denotes the integer Key or Value tensor, $S_{\text{asym}}$ and $Z = \min(\mathbf{X}_{\text{BF16}})$ denote the scaling factor and the zero point respectively, $b$ denotes the quantization bit-width, $\lfloor \cdot \rceil$ denotes the rounding function.

Specifically, MKLV (Hariri et al., 2025) discovers that the sensitivity of Key and Value tensors are quite different, with the Key tensors being more sensitive to quantization than the Value tensors. Therefore, MKLV simply assigns a higher bit-width to Key tensors and a lower bit-width to Value tensors. WKVQuant (Yue et al., 2024) proposes to change the data flow of the previous KV Cache quantization by using the unquantized current Key and Value to calculate the attention operator, and then quantize the current Key and Value. SKVQ (Duanmu et al., 2024) further improves the WKVQuant by using a sliding window that stores the most recent 128 Key and Value features in floating-point format to reduce the cumulative quantization error. MiKV (Yang et al., 2024) is inspired by H2O (Zhang et al., 2023) to use the heavy-hitter oracle to discover the important tokens in a higher bit-width and quantize the rest of the unimportant tokens into a lower bit-width. KIVI (Liu et al., 2024c) discovers that the Value tensors are much flatter than Key tensors, and the outliers in Key tensors typically appear in certain channels. To this end, KIVI utilizes per-channel quantization for Key Cache and per-token quantization for Value Cache in a group-wise manner to reduce the quantization error. RotateKV (Su et al., 2025) combines the channel-wise equalization and the rotation-based equalization with Hadamard matrices to further reduce the quantization error.

In this paper, we adopt effective strategies from prior work, such as storing the first token in INT16 and using a sliding window for recent tokens. To further reduce quantization errors, we propose two improvements: (1) Progressive Quantization – initially store KV cache in higher precision and gradually lower the bit-width as memory memory becomes saturated; (2) Block-wise Memory Allocation – allocate more memory to sensitive transformer blocks when capacity allows, thereby preserving performance.

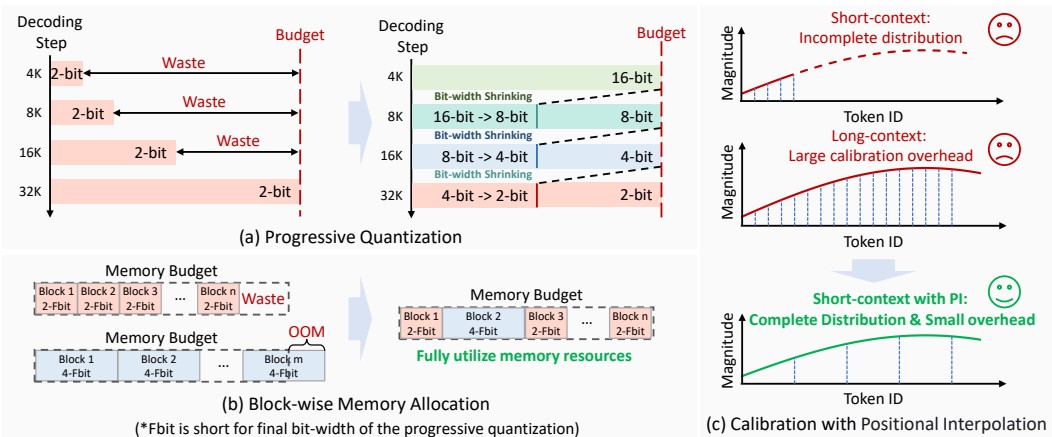

Figure 1: Method Overview. (a) Progressive quantization: we progressively shrink the bit-width of KV Cache to fully utilize the memory budget. (b) Block-wise memory allocation: we allocate a higher bit-width to those transformer blocks with higher sensitivity. (c) Calibration with Positional Interpolation to approximate the distribution of long-context data with short-context data.

## 3 METHOD

### 3.1 PROGRESSIVE QUANTIZATION

As discussed in Section 2.2, existing post-training KV Cache quantization methods quantize at every decoding step, causing large cumulative errors. A sliding window with high-precision cache allevi- ates this, but very low bit-widths (e.g., 2-bit) still lead to severe accuracy loss in long-CoT tasks. **We show that existing KV cache quantization methods underutilize the memory budget and miss opportunities to reduce cumulative errors.** As illustrated in the left panel of Figure 1(a), SOTA methods store 2-bit KV Cache at every decoding step, causing substantial memory waste when the budget is not fully used.

To address the above issue, we propose a progressive quantization strategy to make full use of the memory resources by gradually shrinking the bit-width of the KV Cache, thereby significantly reducing the cumulative quantization error. *For each transformer block, we use "Fbit" to represent the final bit-width of the progressive quantization process.* In this case, we can easily calculate the memory budget for each transformer block based on the maximum context length of the target long- CoT LLM. As shown in Figure 1(a) right, the Fbit in this example is 2-bit and the maximum context length is 32K. During generation, we initially store the KV Cache in 16-bit format to alleviate the large cumulative quantization error. **Once the memory budget is fully utilized**, we apply a bit- width shrinking strategy to accommodate more tokens by progressively reducing the bit-width of the existing KV Cache. Specifically, we use powers of two for quantization bit-widths, gradually decreasing them in the order of 16, 8, 4, and 2 bits.

In addition, for the bit-width shrinking strategy, we design an "**Equivalent Right Shift**" strategy that is mathematically equivalent to de-quantizing the $2b$-bit KV Cache and then quantizing it to $b$-bit. Here, $b$ can be 8, 4, or 2, corresponding to shrinking the KV Cache from 16-bit to 8-bit, 8-bit to 4-bit, and 4-bit to 2-bit, respectively. Specifically, we formulize the bit-width shrinking strategy by using integer addition and shifting as follows:

$$\mathbf{X}_b = \left((2^{2b} - 2^b + 1)(\mathbf{X}_{2b} + 2^{b-1})\right) >> 3b, \quad (3)$$

where $\mathbf{X}_b$ and $\mathbf{X}_{2b}$ represent the $b$-bit and $2b$-bit tensor respectively. We keep the zero point un- changed ($Z_b = Z_{2b}$) and increase the scaling factor to $S_b = (2^b + 1)S_{2b}$ to preserve the dynamic range of the data distribution. The detailed proof of equivalence for Equation (3) is shown in Sec- tion D. Furthermore, we compare the effect of three different bit-width shrinking strategies and show that the "Equivalent Right Shift" strategy achieves better performance, as detailed in Section 4.4.1.

### 3.2 BLOCK-WISE MEMORY ALLOCATION

Existing KV Cache quantization methods typically apply a uniform bit-width across all transformer blocks, which may not fully utilize the memory resources of the target hardwares. As shown in

Figure 1(b) left, in this example, the target hardware has sufficient memory to store the KV Cache uniformly in 2-Fbit format, leaving a proportion of wasted memory. However, switching to a uniform 4-Fbit format may exceed the memory limit and trigger an out-of-memory error. Therefore, using a uniform bit-width for KV Cache may not fully utilize the available memory across different scenarios with varying memory resources.

To fully utilize the memory resource in different scenarios for better performance, we propose a block-wise memory allocation strategy to assign a higher bit-width for more sensitive blocks. Inspired by existing mixed-precision quantization methods (Li et al., 2023; Zhao et al., 2024), we employ a first-order Taylor approximation to estimate the sensitivity of the model output to perturbations in the Key Cache and Value Cache. Here, we take the Key Cache as an example:

$$\mathcal{L}(Q_b(\mathbf{K}_i)) \approx \mathcal{L}(\mathbf{K}) + \mathbf{G}_{\mathbf{K}_i} \odot (\mathbf{K}_i - Q_b(\mathbf{K}_i)), \tag{4}$$

where $\mathcal{L}$ is the loss function, $i$ represents the $i$-th transformer block, $\mathbf{K}_i$ is the Key Cache, $Q_b(\cdot)$ is the $b$-bit quantization function, $\mathbf{G}_{\mathbf{K}_i}$ is the gradients of the loss function with respect to the $\mathbf{K}_i$, $\odot$ is the element-wise multiplication operator. The Value Cache follows a similar formulation.

To minimize the effect of KV Cache quantization in each transformer block, we aim to minimize the following sensitivity term:

$$s_{i,b} = \|\mathbf{G}_{\mathbf{K}_i} \odot (\mathbf{K}_i - Q_b(\mathbf{K}_i))\|_1 + \|\mathbf{G}_{\mathbf{V}_i} \odot (\mathbf{V}_i - Q_b(\mathbf{V}_i))\|_1, \tag{5}$$

where $s_{i,b}$ denotes the sensitivity of the KV Cache in the $i$-th transformer block to $b$-bit quantization.

Taking into account the sensitivity of all transformer blocks, our goal is to assign an appropriate bit-width to each block to minimize the impact on the loss function, subject to a given memory budget. To this end, we formulate the block-wise bit-width allocation as the following Integer Programming problem:

$$\underset{x_{i,b}}{\arg\min} \ \sum_i^N \sum_b x_{i,b} \cdot s_{i,b}, \tag{6}$$

$$\sum_b x_{i,b} = 1, \sum_i^N \sum_b x_{i,b} \cdot (Mem(Q_b(\mathbf{K}_i)) + Mem(Q_b(\mathbf{V}_i))) \leq \mathcal{M}, \tag{7}$$

$$x_{i,b} \in \{0,1\}, b \in B, \tag{8}$$

where $N$ is the number of transformer blocks, $Mem(\cdot)$ is the function to calculate the memory usage of the quantized KV Cache, $\mathcal{M}$ is the memory budget for the KV Cache of all the transformer blocks, $x_{i,b}$ is the one-hot vector that indicates the bit-width choice $b$ of the $i$-th block, and $B$ is the optional bit-width set, detailed in Section 4.1.3. The proposed Integer Programming problem can be effectively solved by CVXPY (Diamond & Boyd, 2016) within a few seconds.

## 3.3 Calibration with Positional Interpolation

Previous studies have observed that the Key Cache of LLMs contains outliers in certain channels, which significantly increases the quantization error. Approaches such as QServe (Lin* et al., 2024) address this issue by introducing a channel-wise reparameterization method to transfer the outliers in Key tensors into Query tensors:

$$\mathbf{P} = (\mathbf{Q}\mathbf{\Lambda}) \cdot Q\left((\mathbf{K}\mathbf{\Lambda}^{-1})^T\right), \mathbf{\Lambda} = diag(\lambda_i), \tag{9}$$

where $i$ is the channel index, $\lambda_i$ is the reparameterization factor of the $i$-th channel, and $Q(\cdot)$ is the quantization function. Generally, $\lambda_i$ is calibrated using a small dataset of sequences with a typical length of 512 tokens, which is much shorter than the maximum output length of 32K tokens. The calibration process follows Equation (10):

$$\lambda_i = \left(\max_m K_{m,i}\right)^\alpha, \tag{10}$$

where $m$ is the token position index, and $\alpha$ is the parameter to adjust the strength of outlier transfer, which can be set as a fixed number or obtained by grid search (Lin et al., 2024).

However, applying the above reparameterization technique to long-CoT LLMs using short calibration data (e.g., 512) may fail to accurately capture the distribution of the Key Cache. This limitation arises because Rotary Positional Embedding (RoPE) (Su et al., 2024) is used to inject positional information into the Key Cache, which introduces periodic variations across different channels:

$$\begin{bmatrix} \widetilde{K}_{m,i} \\ \widetilde{K}_{m,i+\frac{d}{2}} \end{bmatrix} = \begin{bmatrix} \cos m\theta_i & -\sin m\theta_i \\ \sin m\theta_i & \cos m\theta_i \end{bmatrix} \begin{bmatrix} K_{m,i} \\ K_{m,i+\frac{d}{2}} \end{bmatrix} = \sqrt{K_{m,i}^2 + K_{m,i+\frac{d}{2}}^2} \begin{bmatrix} \cos(m\theta_i + \varphi) \\ \sin(m\theta_i + \varphi) \end{bmatrix}, \quad (11)$$

where $K$ and $\widetilde{K}$ denote the Keys before and after RoPE respectively, $d$ is the hidden dimension of each attention head, and $\theta_i$ denotes the rotary frequency of channel $i$ and $i + d/2$. Since $\theta_i = \theta^{-2i/d}$ decreases with increasing $i$, the frequency of the sine curve is extremely low in channels with indices near $d/2$ and $d$. For example, in the DeepSeek-R1-Distill-Qwen-7B, the lowest frequency sine curve has a period of up to 54,410 tokens. Therefore, when using short sequences of 512 tokens for calibration, as shown in Figure 1(c) top, we cannot obtain the reparameterization factor that can completely reflect the sine-like data distribution.

Directly increasing the length of calibration data significantly increases both latency and memory costs due to the $O(N^2)$ complexity of the self-attention operator. Instead, we embed long-context positional information into short calibration data by leveraging positional interpolation (Chen et al., 2023). Specifically, we multiply a position scaling factor $s$ to the position index $m$ in the rotary matrix of RoPE for positional interpolation, as shown below:

$$\begin{bmatrix} \widetilde{K}_{m,i} \\ \widetilde{K}_{m,i+\frac{d}{2}} \end{bmatrix} = \begin{bmatrix} \cos(s \cdot m\theta_i) & -\sin(s \cdot m\theta_i) \\ \sin(s \cdot m\theta_i) & \cos(s \cdot m\theta_i) \end{bmatrix} \begin{bmatrix} K_{m,i} \\ K_{m,i+\frac{d}{2}} \end{bmatrix} = \sqrt{K_{m,i}^2 + K_{m,i+\frac{d}{2}}^2} \begin{bmatrix} \cos(s \cdot m\theta_i + \varphi) \\ \sin(s \cdot m\theta_i + \varphi) \end{bmatrix}. \tag{12}$$

As shown in Figure 1(c) bottom, by applying positional interpolation, we can increase the largest positional index by $s\times$ without additional computation and memory overhead.

### 3.4 METHOD PIPELINE

In this paper, the proposed PM-KVQ combines the above three techniques to achieve better long-CoT performance with low bit-width KV Cache quantization. (1) Before the inference process, we first profile the sensitivity of each transformer block based on the calibration dataset, detailed in Section 4.1.1, and solve the Integer Programming problem to set the proper Fbit for each transformer block, as discussed in Section 3.2. Then, we apply the channel-wise reparameterization technique by using the calibration dataset with positional interpolation, as detailed in Section 3.3. (2) During the inference process, we apply progressive quantization to the KV Cache by gradually lowering the bit-width from 16-bit to the allocated Fbit, as shown in Section 3.1.

## 4 EXPERIMENTS

### 4.1 EXPERIMENTAL SETUPS

#### 4.1.1 DATASETS

**For the calibration dataset**, we use the arXiv subset of RedPajama (Weber et al., 2024) as calibration dataset. This subset consists of academic papers, containing mathematical formulas and reasoning process. We randomly select 512 samples, each with a length of 2,048 tokens, for calibration. For positional interpolation, we set $s = 4$ in Equation (12), which means we embed an 8,192 context length to 2,048 tokens. We set $\alpha$ in Equation (10) by grid searching over [0,1] for the optimal $\alpha$ that minimizes the reconstruction loss of the self-attention operator with a grid size of 20.

**For performance evaluation**, we mainly focus on evaluating the long-CoT LLMs on the mathematical reasoning and code generation benchmarks with **long generation contexts (>16K)**. For mathematical reasoning, we use the AIME-2024/2025 (AIME, 2025) and CMIMC-2024 (CMIMC, 2025) datasets. For competition-level code generation, we select coding problems released between January 1, 2025, and April 6, 2025, from LiveCodeBench (Jain et al., 2024). Besides, as illustrated in Section C.2, we also evaluate the proposed PM-KVQ on the IFEval (Zhou et al., 2023) dataset with **short generation contexts (∼1K)** to demonstrate its strong generalizability across different

context lengths. We sample 16 responses for each mathematical problem, 4 responses for each code generation problem, and 1 response for each instruction following problem, using a temperature of 0.6, top-p of 0.95, and a maximum output length of 32,768 tokens.

### 4.1.2 BASELINES AND MODEL CHOICE

**For baselines**, we compare PM-KVQ with SOTA KV Cache quantization methods, including the uniform bit-width methods RotateKV (Su et al., 2025), KIVI (Liu et al., 2024c), and mixed-precision quantization method MiKV (Yang et al., 2024), which retains the KV Cache of heavy hitters in BF16 format and uses low bit-width for other tokens. We also compare PM-KVQ with KVTuner (Li et al., 2025) in Section C.4. Similar to KIVI, PM-KVQ stores the KV Cache for the first and most recent 128 tokens in INT16 format to mitigate performance degradation. All model weights in our experiments are in BF16 format.

**For model choices**, we evaluate the different quantization methods above on the Deepseek-R1-Distill (Guo et al., 2025) series as well as the QwQ-32B model (Team, 2025). Specifically, the Deepseek-R1-Distill series is an LLM family distilled from DeepSeek-R1. We choose Deepseek-R1-Distill-Qwen-7B/14B/32B and Deepseek-R1-Distill-LLaMA-8B/70B, ranging from 7B to 70B.

### 4.1.3 BIT-WIDTH AND BATCH SIZE SETUPS

**For the bit-width settings**, to demonstrate the effectiveness of the proposed PM-KVQ, we select quantization bit-widths that lead to significant performance degradation when using baseline methods for each long-CoT LLM. Specifically, we use 4-bit for DeepSeek-LLaMA-8B and 2-bit for other LLMs. Notably, the bit-width for the proposed PM-KVQ stands for the Fbit, as discussed in Section 3.1. In addition, for the optional bit-width set $B$ in Section 3.2, we use $B = \{4, 8\}$ for DeepSeek-LLaMA-8B, and $B = \{2, 4\}$ for other long-CoT LLMs. We use asymmetric group-wise quantization for KV Cache with a group size of 128, as shown in Equation (1). All of the performance results are conducted with fake quantization on an $8 \times$A100-80G GPU server.

**For the batch size setups**, we assign a target GPU with different memory resources for different LLMs to show the memory constraints in real-world scenarios, as shown in Table 2. On the one hand, to demonstrate the effectiveness of progressive quantization, we set the batch size for each LLM such that all methods can fully utilize the memory resources of the target GPU. Specifically, we use a batch size of 8 for LLaMA-8B with a 4-bit KV Cache, 40 for Qwen-7B with a 2-bit KV Cache, and 16 for the other LLMs, as shown in Table 2. On the other hand, to evaluate the effectiveness of block-wise memory allocation, we use smaller batch sizes to allocate more memory per instance, ensuring that higher bit-widths cannot be directly used under the same constraints. In this setting, we use a batch size of 6 for LLaMA-8B with a 4-bit KV Cache, 32 for Qwen-7B with a 2-bit KV Cache, and 12 for the remaining LLMs, as also shown in Table 2. Results across more target hardwares can be found in Section C.5.

## 4.2 MAIN RESULTS

As illustrated in Table 2, for long-CoT LLMs smaller than 10B, we compare PM-KVQ with RotateKV, MiKV, and KIVI. For the 2-bit DeepSeek-R1-Distill-Qwen-7B, applying RotateKV or MiKV causes the model unable to generate meaningful responses. The SOTA method KIVI also suffers from significant performance loss by up to 9%. PM-KVQ outperforms KIVI by up to 8% when applying uniform Fbit for each transformer block (batch size = 40). When the batch size is reduced to 32, each sample receives a larger memory budget. However, this budget is still insufficient to apply uniform 4-bit quantization across all blocks. As a result, KIVI is constrained to 2-bit quantization, underutilizing the available memory. In contrast, PM-KVQ leverages block-wise memory allocation to better utilize the larger memory, achieving an additional performance gain of up to 0.84%. For the 4-bit DeepSeek-R1-Distill-LLaMA-8B, PM-KVQ surpasses the SOTA methods by up to 6.5% on AIME-2024, and even achieve better performance than the original LLM on mathematical benchmarks. Besides, for LLMs smaller than 10B, the average voting accuracy of PM-KVQ exceeds KIVI by up to 15.56%, demonstrating greater stability of the proposed method. We also compare PM-KVQ with KIVI of different bit-widths in Section C.3.

For larger long-CoT LLMs from 10B to 32B, we only compare the proposed PM-KVQ with KIVI because MiKV and RotateKV fail to generate meaningful information under 2-bit quantization, as

Table 2: Main results of long-CoT Language Models on reasoning-related benchmarks with SOTA KV Cache quantization methods. "BS" is short for "batch size".

| Models (Target GPU) | Quantization Methods | Bit-width (K-V) | AIME-2024 pass@1 | AIME-2024 Voting | AIME-2025 pass@1 | AIME-2025 Voting | CMIMC-2024 pass@1 | CMIMC-2024 Voting | LiveCode pass@1 |
|---|---|---|---|---|---|---|---|---|---|
| DeepSeek-Qwen-7B (1×4090-24G) | - - | 16-16 | $41.04_{\pm6.74}$ | 63.33 | $30.00_{\pm3.33}$ | 36.67 | $27.29_{\pm5.17}$ | 43.33 | $26.29_{\pm1.34}$ |
| | RotateKV (BS=32,40) | 2-2 | $0.00_{\pm0.00}$ | 0.00 | $0.00_{\pm0.00}$ | 0.00 | $0.00_{\pm0.00}$ | 0.00 | $0.00_{\pm0.00}$ |
| | MiKV (BS=32) | 2/16-2/16 | $0.00_{\pm0.00}$ | 0.00 | $0.63_{\pm0.02}$ | 3.33 | $2.29_{\pm0.02}$ | 3.33 | $5.86_{\pm0.85}$ |
| | MiKV (BS=40) | 2-2 | $0.00_{\pm0.00}$ | 0.00 | $0.00_{\pm0.00}$ | 0.00 | $0.00_{\pm0.00}$ | 0.00 | $0.00_{\pm0.00}$ |
| | KIVI (BS=32,40) | 2-2 | $32.08_{\pm5.25}$ | 43.33 | $24.58_{\pm3.51}$ | 33.33 | $20.83_{\pm3.63}$ | 23.33 | $19.00_{\pm2.37}$ |
| | PM-KVQ (BS=32) | 2/4-2/4 | $\mathbf{40.21}_{\pm5.71}$ | **66.67** | $\mathbf{28.96}_{\pm4.20}$ | **40.00** | $25.83_{\pm5.20}$ | **40.00** | $\mathbf{24.71}_{\pm1.48}$ |
| | PM-KVQ (BS=40) | 2-2 | $40.00_{\pm5.40}$ | 60.00 | $28.12_{\pm4.71}$ | 33.33 | $\mathbf{26.46}_{\pm4.64}$ | **40.00** | $24.57_{\pm1.42}$ |
| DeepSeek-LLaMA-8B (1×4090-24G) | - - | 16-16 | $44.17_{\pm4.49}$ | 66.67 | $30.63_{\pm6.58}$ | 50.00 | $26.67_{\pm4.41}$ | 36.67 | $32.14_{\pm1.99}$ |
| | RotateKV (BS=6,8) | 4-4 | $42.92_{\pm3.89}$ | 66.67 | $27.29_{\pm6.48}$ | 40.00 | $26.46_{\pm5.33}$ | 30.00 | $\mathbf{32.00}_{\pm1.56}$ |
| | MiKV (BS=6) | 4/16-4/16 | $35.63_{\pm7.14}$ | 66.67 | $24.79_{\pm3.72}$ | 36.67 | $25.21_{\pm3.53}$ | 33.33 | $27.00_{\pm1.30}$ |
| | MiKV (BS=8) | 4-4 | $41.67_{\pm6.56}$ | 60.00 | $26.46_{\pm7.02}$ | 43.33 | $22.92_{\pm4.84}$ | 26.67 | $29.71_{\pm1.67}$ |
| | KIVI (BS=6,8) | 4-4 | $41.25_{\pm6.65}$ | 60.00 | $27.92_{\pm4.70}$ | 46.67 | $26.25_{\pm4.98}$ | 36.67 | $30.29_{\pm1.76}$ |
| | PM-KVQ (BS=6) | 4/8-4/8 | $\mathbf{47.71}_{\pm6.84}$ | **73.33** | $\mathbf{31.25}_{\pm5.64}$ | 50.00 | $28.13_{\pm4.08}$ | 36.67 | $31.71_{\pm0.86}$ |
| | PM-KVQ (BS=8) | 4-4 | $43.33_{\pm5.57}$ | 63.33 | $\mathbf{31.25}_{\pm5.64}$ | 50.00 | $\mathbf{28.96}_{\pm5.10}$ | **40.00** | $31.57_{\pm1.17}$ |
| DeepSeek-Qwen-14B (1×A100-40G) | - - | 16-16 | $68.13_{\pm7.26}$ | 80.00 | $50.00_{\pm5.77}$ | 60.00 | $49.58_{\pm4.84}$ | 66.67 | $45.71_{\pm1.34}$ |
| | KIVI (BS=12,16) | 2-2 | $48.13_{\pm4.85}$ | 70.00 | $33.96_{\pm3.17}$ | 43.33 | $27.71_{\pm3.67}$ | 33.33 | $34.43_{\pm3.11}$ |
| | PM-KVQ (BS=12) | 2/4-2/4 | $\mathbf{67.71}_{\pm6.94}$ | 80.00 | $\mathbf{46.67}_{\pm7.36}$ | 60.00 | $\mathbf{47.71}_{\pm4.20}$ | 60.00 | $\mathbf{42.14}_{\pm0.95}$ |
| | PM-KVQ (BS=16) | 2-2 | $63.33_{\pm4.08}$ | **83.33** | $42.08_{\pm6.55}$ | 60.00 | $46.67_{\pm5.27}$ | **70.00** | $41.86_{\pm1.78}$ |
| DeepSeek-Qwen-32B (1×A100-80G) | - - | 16-16 | $72.08_{\pm4.39}$ | 86.67 | $53.12_{\pm5.71}$ | 66.67 | $52.50_{\pm5.71}$ | 70.00 | $46.86_{\pm2.18}$ |
| | KIVI (BS=12,16) | 2-2 | $63.96_{\pm6.89}$ | **83.33** | $45.42_{\pm5.38}$ | 60.00 | $40.63_{\pm5.17}$ | 56.67 | $40.43_{\pm1.10}$ |
| | PM-KVQ (BS=12) | 2/4-2/4 | $\mathbf{69.17}_{\pm5.95}$ | **83.33** | $48.54_{\pm5.89}$ | 60.00 | $\mathbf{51.25}_{\pm4.70}$ | 66.67 | $\mathbf{43.57}_{\pm1.64}$ |
| | PM-KVQ (BS=16) | 2-2 | $67.29_{\pm4.89}$ | **83.33** | $\mathbf{48.96}_{\pm7.33}$ | 63.33 | $50.42_{\pm7.16}$ | **73.33** | $\mathbf{43.57}_{\pm0.62}$ |
| QwQ-32B (1×A100-80G) | - - | 16-16 | $78.54_{\pm4.85}$ | 86.67 | $67.71_{\pm3.48}$ | 76.67 | $71.25_{\pm3.51}$ | 80.00 | $54.71_{\pm0.74}$ |
| | KIVI (BS=12,16) | 2-2 | $61.25_{\pm5.51}$ | 76.67 | $\mathbf{51.67}_{\pm5.27}$ | 63.33 | $48.33_{\pm5.77}$ | 63.33 | $41.86_{\pm1.21}$ |
| | PM-KVQ (BS=12) | 2/4-2/4 | $66.46_{\pm3.81}$ | **80.00** | $49.58_{\pm4.39}$ | 63.33 | $54.58_{\pm5.12}$ | 66.67 | $\mathbf{45.14}_{\pm0.70}$ |
| | PM-KVQ (BS=16) | 2-2 | $\mathbf{67.29}_{\pm3.38}$ | 76.67 | $49.79_{\pm6.29}$ | **70.00** | $\mathbf{56.67}_{\pm3.91}$ | **73.33** | $44.57_{\pm0.40}$ |

discovered in the 2-bit DeepSeek-R1-Distill-Qwen-7B. As shown in Table 2, PM-KVQ also demonstrates superior performance compared to KIVI, improving average pass@1 and voting accuracy by up to 15.00% and 17.78% on various LLMs. Especially, for the DeepSeek-R1-Distill-Qwen-14B, KIVI causes a performance degradation of 21.87% on CMIMC-2024, whereas PM-KVQ has a significantly lower degradation of only 1.87% and 2.91% under batch sizes of 16 and 12, respectively.

For the 70B-level long-CoT LLM, we evaluate the 2-bit DeepSeek-R1-Distill-LLaMA-70B model on the AIME-2024 benchmark. The original 16-bit model achieves a pass@1 of 69.14%. When the KV Cache is quantized to 2-bit using KIVI, the pass@1 drops significantly to 51.88%. In contrast, the proposed PM-KVQ enables the 2-bit model to achieve a much higher pass@1 of 64.79% under both batch sizes of 12 and 16, outperforming the KIVI baseline by 12.91%.

## 4.3 EFFICIENCY ANALYSIS

We evaluate 7B and 32B long-CoT LLMs on an A100-80G GPU, comparing the throughput of PM-KVQ (Fbit=2) against the original 16-bit LLMs and the 2-bit KIVI baseline. We adopt the official settings of KIVI (Liu et al., 2024c), using its inference engine and 4/2-bit CUDA kernels for efficiency evaluation. Besides, we implement 16/8-bit CUDA kernels and bit-width shrinking kernels to support PM-KVQ. To fully utilize the A100-80G memory, we set the batch sizes of the original 7B and 32B models to 18 and 1, respectively, while the quantized models allow larger batch sizes of 110 and 4.

Table 3: The throughput (in tokens/s) across different quantization methods and output lengths.

| Model | Quantization Method | Output Length 16K | Output Length 24K | Output Length 32K |
|---|---|---|---|---|
| DeepSeek-Qwen-7B | – | 101.40 | 65.69 | 52.06 |
| | KIVI | 506.72 | 352.44 | 284.10 |
| | PM-KVQ | 424.72 | 323.48 | 269.51 |
| DeepSeek-Qwen-32B | – | 12.34 | 10.35 | 8.92 |
| | KIVI | 35.65 | 33.28 | 31.59 |
| | PM-KVQ | 33.74 | 32.15 | 30.81 |

As shown in Table 3, across different model sizes and output lengths, PM-KVQ achieves a 2.73–5.18× throughput improvement over the original 16-bit LLMs. Compared with KIVI, the throughput of PM-KVQ is at a similar level, with a slight reduction primarily due to the use of higher bit-widths during inference. Notably, the overhead of bit-width shrinking is negligible, as it is trig-

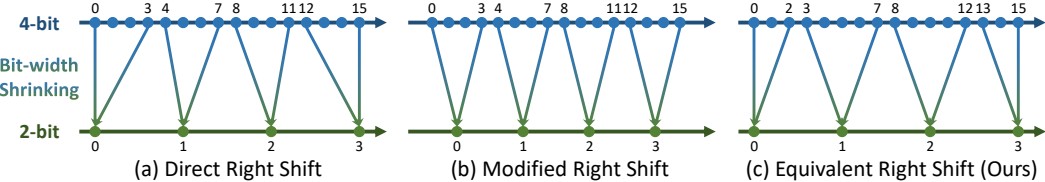

Figure 2: Different bit-width shrinking strategies when the bit-width is reduced from 4-bit to 2-bit.

gered only when memory is fully utilized. **Overall, PM-KVQ incurs a throughput degradation of 2.45–16.18% compared to KIVI but achieves a substantial relative accuracy improvement of 10.57–23.48%.** To further evaluate the efficiency of the quantization procedure, we measure the latency of block-wise memory allocation and calibration with positional interpolation. As shown in Section C.7, both the 7B and 32B LLMs complete these procedures within one hour using PM-KVQ.

## 4.4 ABLATION STUDIES

In this section, we conduct detailed ablation studies to show the effect of bit-wise shrinking strategies introduced in Section 3.1, the sensitivity of different transformer blocks discussed in Section 3.2, and the effectiveness of the positional interpolation discussed in Section 3.3.

### 4.4.1 THE EFFECT OF BIT-WIDTH SHRINKING STRATEGIES

Table 4: Ablation results of different bit-width shrinking strategies.

| Model | Bit-width Shrinking Strategy | Bit-width (K-V) | AIME-2024 | |
|---|---|---|---|---|
| | | | pass@1 | Voting |
| DeepSeek-LLaMA-8B | - - | 16-16 | 44.17 | 66.67 |
| | Direct Right Shift | 4-4 | 12.08 | 23.33 |
| | Modified Right Shift | 4-4 | 28.75 | 46.67 |
| | Equivalent Right Shift (Ours) | 4-4 | 38.33 | 66.67 |

We compare three different bit-width shrinking strategies for reducing the KV Cache from $2b$-bit to $b$-bit. Specifically, $b$ can be 8, 4, or 2, corresponding to shrinking the KV Cache from 16-bit to 8-bit, 8-bit to 4-bit, and 4-bit to 2-bit, respectively.

(1) **Direct Right Shift**: By directly right-shifting by $b$ bits, only the higher $b$ bits of the original $2b$-bit value are retained. As shown in Figure 2 (a), to preserve the dynamic range of the quantized values, we keep the zero point unchanged ($Z_b = Z_{2b}$) and increase the scaling factor to $S_b = (2^b + 1)S_{2b}$ to compensate for the magnitude reduction caused by the right-shift operation.

(2) **Modified Right Shift**: This strategy also uses $b$-bit right shifting strategy to perform the bit-width shrinking. However, instead of keeping the dynamic range unchanged, this strategy aims to ensure that quantization levels sharing the same upper $b$ bits before the shift can have their mean values directly mapped to the lower bit-width representation, as demonstrated in Figure 2 (b). To achieve this, we change the scaling factor by $S_b = 2^b \cdot S_{2b}$ and zero point by $Z_b = Z_{2b} + \frac{1}{2}(S_b - S_{2b})$.

(3) **Equivalent Right Shift (in Section 3.1)**: As shown in Figure 2 (c), this strategy is equivalent to directly de-quantizing the $2b$-bit KV Cache and then quantizing it to $b$-bit.

We evaluate the above three bit-width shrinking strategies on the AIME-2024 benchmark with DeepSeek-R1-Distill-LLaMA-8B. As shown in Table 4, both the Direct Right Shift and Modified Right Shift strategies result in significant performance degradation, reducing the pass@1 by 32.09% and 15.42%, respectively. In contrast, the Equivalent Right Shift demonstrates a notable improvement over the other two strategies, increasing the pass@1 by 26.25% and 9.58%, and maintaining a lossless voting accuracy. Therefore, we adopt the Equivalent Right Shift strategy in PM-KVQ.

### 4.4.2 THE SENSITIVITY OF DIFFERENT TRANSFORMER BLOCKS

We analyze the sensitivity and the memory allocation results across different models. For models with parameter size less than 10B, as shown in Figure 3, we observe that the deeper blocks tend to be more sensitive to quantization and receive a larger memory budget for the KV Cache. In addition,

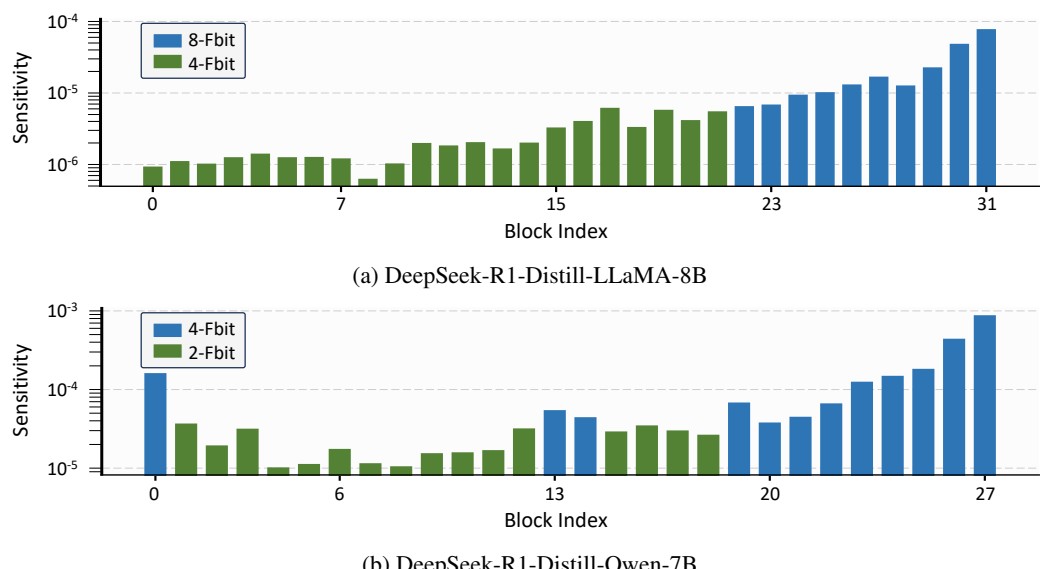

(a) DeepSeek-R1-Distill-LLaMA-8B

(b) DeepSeek-R1-Distill-Qwen-7B

Figure 3: Sensitivity to quantization of KV Cache in different transformer blocks. Different colors represents different memory budgets.

in the DeepSeek-R1-Distill-Qwen-7B model, the first block is more sensitive than the other shallow blocks. Our memory allocation strategy accurately captures this feature, assigning a higher memory budget to the first block accordingly. The block sensitivity of larger models is detailed in Section C.1.

### 4.4.3 THE EFFECT OF POSITIONAL INTERPOLATION

We evaluate the long-CoT performance across varying lengths of calibration data and position scaling factor $s$. We utilize the DeepSeek-R1-Distill-LLaMA-8B to generate four responses for each problem in the AIME-2024-I dataset. As shown in Table 5, when the calibration sequence length is set to 2,048, applying positional interpolation with $s = 4$ improves pass@1 by 1.66% compared to not using positional interpolation, achieving accuracy comparable to that obtained using calibration sequences of 8,192 tokens. We also observe that when $s$ increases to 16, positional interpolation may lead to performance degradation. This indicates that the computational savings of positional interpolation are not unlimited, and overly aggressive scaling can indeed performance drop.

Table 5: Ablation results of different calibration sequence lengths and position scaling factors.

| Model | Calibration Sequence Length | Position Scaling Factor | Effective Length | AIME-2024-I | |
| --- | --- | --- | --- | --- | --- |
| | | | | pass@1 | Voting |
| DeepSeek-LLaMA-8B | 2,048 | 1 | 2,048 | 46.67 | 60.00 |
| | 2,048 | 4 | 8,192 | 48.33 | 60.00 |
| | 2,048 | 16 | 32,768 | 46.67 | 53.33 |
| | 8,192 | 1 | 8,192 | 48.33 | 60.00 |

## 5 CONCLUSION

In this paper, we introduce Progressive Mixed-precision KV Cache Quantization (PM-KVQ), a post-training KV Cache quantization method designed for long-CoT LLMs. To reduce the large cumulative error caused by uniform bit-width quantization, we design progressive quantization and block-wise memory allocation techniques. To increase the effective calibration length without incurring additional overhead, we propose a new calibration strategy with positional interpolation. Extensive experiments and ablation studies demonstrate the effectiveness of the proposed PM-KVQ and each proposed technique. Overall, the proposed PM-KVQ significantly outperforms SOTA baselines by up to 8% on reasoning-related mathematics and coding benchmarks and achieves 2.73–5.18× throughput compared to the original 16-bit LLMs.

ACKNOWLEDGEMENT

This work was supported by National Natural Science Foundation of China (No. 62325405, 62104128, U19B2019, U21B2031, 61832007, 62204164, 92364201), Tsinghua EE Xilinx AI Research Fund, and Beijing National Research Center for Information Science and Technology (BN-Rist). We thank for all the support from Infinigence-AI.

ETHICS STATEMENT

This work focuses on reducing the substantial overhead caused by the linearly growing KV cache in long-context processing through KV Cache quantization. On the one hand, the proposed PM-KVQ better preserves model accuracy after low-precision KV cache quantization, making it more accessible for cost-constrained institutions, individuals, and application scenarios. On the other hand, as a lossy compression technique, quantization can introduce distribution shifts and performance degradation, potentially leading to increased hallucinations or instruction-following failures. Therefore, additional caution and oversight are required during deployment.

REPRODICIBILITY STATEMENT

We describe the calibration and evaluation datasets, as well as the data processing procedures, in Section 4.1.1. All datasets and models used in our experiments are publicly available. Detailed information on the quantization bit-widths and batch sizes used for each long-CoT LLM is also provided in Section 4.1.3. To facilitate reproducibility, we also release our source code along with detailed guidelines.

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

## A  THE USE OF LARGE LANGUAGE MODELS (LLMS)

In this paper, LLMs are only used to assist in polishing the writing of this paper. The technical content, experiments, and conclusions are entirely conceived and conducted by the authors.

## B  ADDITIONAL DETAILS OF EVALUATION

### B.1  INTRODUCTION OF DATASETS

**American Invitational Mathematics Examination (AIME)** (AIME, 2025) is a mathematics competition for high school students. It contains 30 challenging problems each year, designed to assess mathematical problem-solving skills across various topics, including algebra, combinatorics, geometry, number theory, and other subjects covered in high school curricula.

**Carnegie Mellon Informatics and Mathematics Competition (CMIMC)** (CMIMC, 2025) is an annual mathematics contest for high school students, hosted by students from Carnegie Mellon University. The competition contains problems of algebra, combinatorics, and geometry, with each category including ten standard problems along with one tiebreaker. Our model evaluation focuses on the standard problem sets.

**LiveCodeBench** (Jain et al., 2024) is an extensive and continuously updated benchmark designed to evaluate the performance of LLMs in coding tasks. It continually gathers new problems from competition platforms. The benchmark encompasses four distinct scenarios: code generation, automated code repair, code execution, and prediction of test outputs. In our experiments, we focus specifically on the code generation scenario.

**IFEval** (Zhou et al., 2023) is a benchmark proposed to systematically evaluate the ability of LLMs to follow natural language instructions. The dataset contains 541 prompts, each annotated with one or more verifiable instruction types such as word-count constraints, keyword frequency, formatting requirements, or prohibitions on certain symbols. These instructions were deliberately designed to be automatically checkable, enabling objective and reproducible evaluation without the need for human annotators.

## C  ADDITIONAL EXPERIMENTS

### C.1  THE SENSITIVITY OF DIFFERENT TRANSFORMER BLOCKS

For models with parameter size over 10B, as shown in Figure 4, KV Cache in deeper blocks tend to be more sensitive than shallower blocks. We also observe that for the Qwen-based models, the first block exhibits a large sensitivity. In particular, the sensitivity of the first block is the largest among the first fifteen blocks in different Qwen-based models. This phenomenon is not observed in the LLaMA-based models.

### C.2  PERFORMANCE IN SHORT-GENERATION-CONTEXT TASKS

To verify the scalability of PM-KVQ to short-generation-context tasks, we evaluate it on IFEval (Zhou et al., 2023), an instruction-following benchmark. We follow the experimental setup described in Section 4.1 and adopt the evaluation metrics provided by OpenCompass (Contributors, 2023). Compared to reasoning benchmarks in Table 2, non-reasoning tasks are less challenging and generally involve much shorter outputs. For instance, the average output length of the DeepSeek-Qwen-7B model is 13,904 tokens on AIME-2024 but only 1,182 tokens on IFEval. As shown in Table 6, although our method is not specifically designed for short-output scenarios, it outperforms KIVI and achieves accuracy comparable to the original 16-bit models.

### C.3  COMPARISON WITH KIVI OF DIFFERENT BIT-WIDTHS

In our experiments in Section 4.2, PM-KVQ occupies more memory than KIVI before the KV cache bit-width is reduced to the final Fbit. However, under uniform Fbit settings, both our approach and KIVI consume the same amount of memory once the KV cache bit-width is reduced to Fbit. In fact,

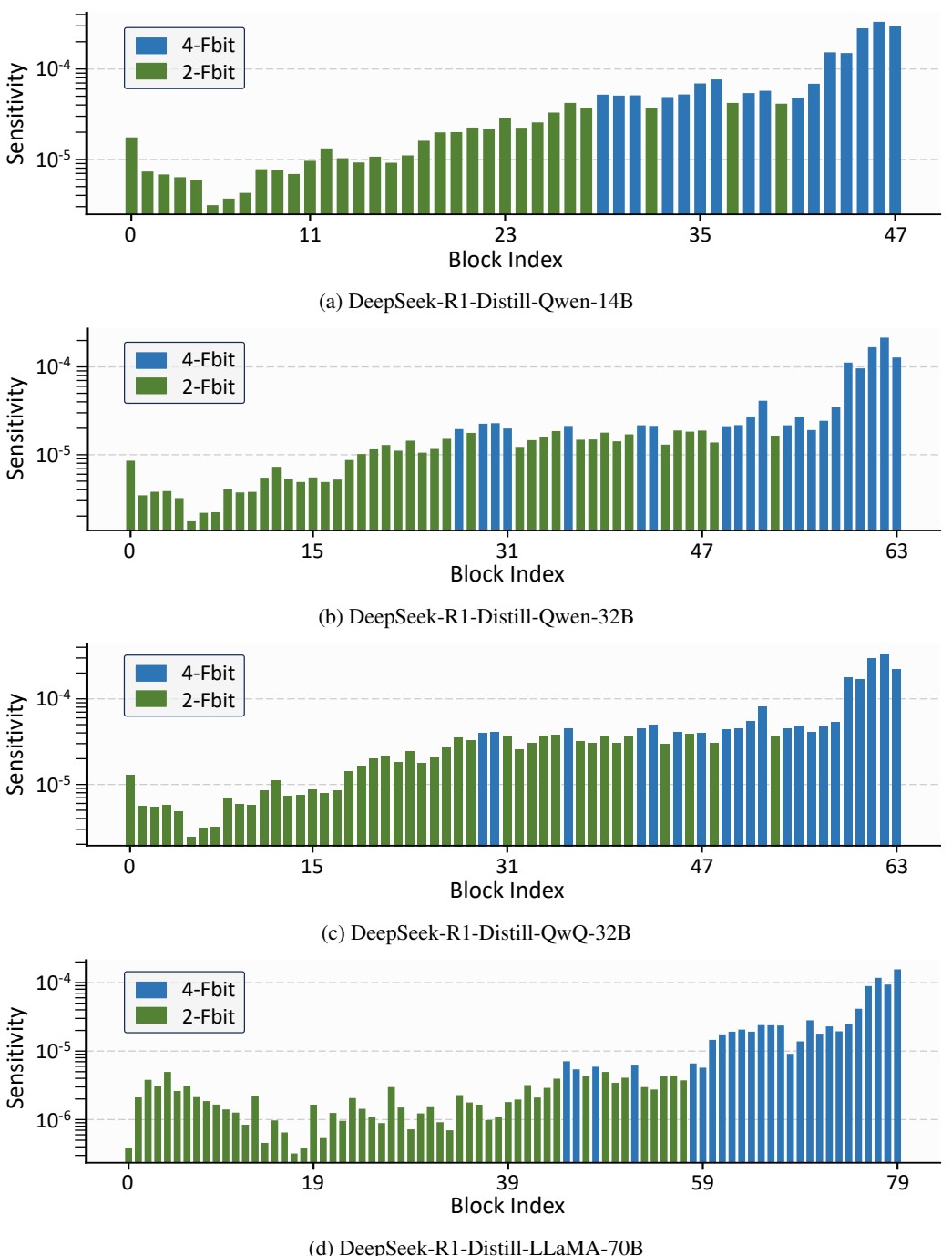

(a) DeepSeek-R1-Distill-Qwen-14B

(b) DeepSeek-R1-Distill-Qwen-32B

(c) DeepSeek-R1-Distill-QwQ-32B

(d) DeepSeek-R1-Distill-LLaMA-70B

Figure 4: Sensitivity to quantization of KV Cache in different transformer blocks. Different colors represents different memory budgets.

Table 6: Results of long-CoT Language Models on non-reasoning benchmarks with SOTA KV Cache quantization methods. "BS" is short for "batch size".

| Models | Quantization | Bit-width | IFEval | | | |
|---|---|---|---|---|---|---|
| (Target GPU) | Methods | (K-V) | Prompt Strict | Prompt Loose | Instruct Strict | Instruct Loose |
| DeepSeek-Qwen-7B (1×4090-24G) | - - | 16-16 | 58.77 | 68.50 | 63.27 | 72.60 |
| | RotateKV (BS=32,40) | 2-2 | 0.00 | 0.00 | 0.00 | 0.00 |
| | MiKV (BS=32) | 2/16-2/16 | 57.30 | 66.17 | 61.18 | 70.06 |
| | MiKV (BS=40) | 2-2 | 0.00 | 0.00 | 0.00 | 0.00 |
| | KIVI (BS=32,40) | 2-2 | 49.29 | 60.31 | 54.57 | 64.57 |
| | PM-KVQ (BS=32) | 2/4-2/4 | 57.35 | 68.03 | **62.09** | 71.65 |
| | PM-KVQ (BS=40) | 2-2 | **57.58** | **68.34** | **62.09** | **71.81** |
| DeepSeek-LLaMA-8B (1×4090-24G) | - - | 16-16 | 57.82 | 68.98 | 61.61 | 71.81 |
| | RotateKV (BS=6,8) | 4-4 | 58.06 | 68.50 | 61.37 | 71.50 |
| | MiKV (BS=6) | 4/16-4/16 | 44.08 | 56.38 | 46.92 | 59.53 |
| | MiKV (BS=8) | 4-4 | 55.69 | 66.61 | 59.95 | 70.39 |
| | KIVI (BS=6,8) | 4-4 | 57.35 | 68.35 | 71.14 | 71.81 |
| | PM-KVQ (BS=6) | 4/8-4/8 | **58.77** | **69.61** | **63.74** | **73.70** |
| | PM-KVQ (BS=8) | 4-4 | 57.58 | 68.19 | 61.14 | 71.34 |
| DeepSeek-Qwen-14B (1×A100-40G) | - - | 16-16 | 70.14 | 78.74 | 74.40 | 81.73 |
| | KIVI (BS=12,16) | 2-2 | 67.54 | 77.01 | 72.04 | 80.47 |
| | PM-KVQ (BS=12) | 2/4-2/4 | **73.70** | **80.47** | **77.73** | **83.46** |
| | PM-KVQ (BS=16) | 2-2 | 73.46 | **80.47** | 77.49 | **83.46** |
| DeepSeek-Qwen-32B (1×A100-80G) | - - | 16-16 | 74.41 | 81.73 | 78.20 | 84.41 |
| | KIVI (BS=12,16) | 2-2 | 72.51 | 79.84 | 76.07 | 82.36 |
| | PM-KVQ (BS=12) | 2/4-2/4 | 75.83 | 83.15 | **78.91** | **85.35** |
| | PM-KVQ (BS=16) | 2-2 | **76.07** | **83.30** | **78.91** | **85.35** |
| QwQ-32B (1×A100-80G) | - - | 16-16 | 82.94 | 88.03 | 86.97 | 90.71 |
| | KIVI (BS=12,16) | 2-2 | 73.22 | 80.00 | 78.67 | 84.09 |
| | PM-KVQ (BS=12) | 2/4-2/4 | **81.99** | **86.77** | **85.55** | **89.45** |
| | PM-KVQ (BS=16) | 2-2 | 81.75 | 86.61 | **85.55** | **89.45** |

Table 7: Results of long-CoT Language Models on reasoning-related benchmarks with KIVI of different bit-widths. "BS" is short for "batch size".

| Models | Quantization | Bit-width | AIME-2024 | | AIME-2025 | |
|---|---|---|---|---|---|---|
| (Target GPU) | Methods | (K-V) | pass@1 | Voting | pass@1 | Voting |
| DeepSeek-Qwen-7B (1×4090-24G) | - - | 16-16 | 41.04 | 63.33 | 30.00 | 36.67 |
| | KIVI (BS=32,40) | 16-16 | 21.88 | 36.67 | 21.67 | 26.67 |
| | KIVI (BS=32,40) | 8-8 | 36.67 | 63.33 | 24.59 | 33.33 |
| | KIVI (BS=32,40) | 4-4 | 39.38 | 63.33 | 26.46 | 36.67 |
| | KIVI (BS=32,40) | 2-2 | 32.08 | 43.33 | 24.58 | 33.33 |
| | PM-KVQ (BS=32) | 2/4-2/4 | **40.21** | **66.67** | **28.96** | **40.00** |
| | PM-KVQ (BS=40) | 2-2 | 40.00 | 60.00 | 28.12 | 33.33 |

under the same GPU and batch size constraints, if KIVI adopts a higher quantization bit-width, it will consume more memory and therefore exhaust the available memory budget before reaching the maximum output length. We evaluate KIVI across different bit-width settings. When KIVI fully utilizes the memory budget, we truncate its output accordingly. As shown in Table 7, our method still outperforms KIVI by 0.83%–18.33%.

## C.4 COMPARISON WITH MIXED-PRECISION QUANTIZATION BASELINES

We conduct additional experiments comparing our method with KVTuner on DeepSeek-R1-Distill-Qwen-7B under the same target GPU and batch size settings. For KVTuner, we adopt per-channel asymmetric quantization for the Key cache and per-token asymmetric quantization for the Value cache. As shown in Table 8, the pass@1 of our method surpasses KVTuner by 2.71-6.04%.

Table 8: Results of long-CoT Language Models on reasoning-related benchmarks with the mixed-precision quantization baseline. "BS" is short for "batch size".

| Models | Quantization | Bit-width | AIME-2024 | | AIME-2025 | |
|---|---|---|---|---|---|---|
| (Target GPU) | Methods | (K-V) | pass@1 | Voting | pass@1 | Voting |
| DeepSeek-Qwen-7B | - - | 16-16 | 41.04 | 63.33 | 30.00 | 36.67 |
| | KIVI (BS=32) | 2-2 | 32.08 | 43.33 | 24.58 | 33.33 |
| | KVTuner (BS=32) | 2/4/8-2/4/8 | 34.17 | 56.67 | 26.25 | 33.33 |
| (1×4090-24G) | PM-KVQ (BS=32) | 2/4-2/4 | **40.21** | **66.67** | **28.96** | **40.00** |

## C.5 Performance across Different Hardware Configurations

When the memory capacity of the target GPU increases, PM-KVQ can either maintain the per-request memory allocation by increasing the batch size or allocate a higher Fbit for each layer, allowing the KV cache to use higher bit-widths. We evaluate our method across different GPU memory capacities by increasing the Fbit accordingly. As shown in Table 9, our method consistently achieves accuracy comparable to the original LLM under varying Fbit settings.

Table 9: Performance of PM-KVQ across different target GPUs. "BS" is short for "batch size".

| Models | Target GPU | Quantization | Bit-width | AIME-2024 | |
|---|---|---|---|---|---|
| | | Methods | (K-V) | pass@1 | Voting |
| DeepSeek-Qwen-7B | 1×4090-24G | PM-KVQ (BS=32) | 2/4-2/4 | 40.21 | 66.67 |
| | | PM-KVQ (BS=40) | 2-2 | 40.00 | 60.00 |
| | 1×A100-40G | PM-KVQ (BS=36) | 4/8-4/8 | 40.63 | 63.33 |
| | | PM-KVQ (BS=58) | 4-4 | 40.83 | 66.67 |
| | 1×A100-80G | PM-KVQ (BS=108) | 8/16-8/16 | 42.50 | 66.67 |
| | | PM-KVQ (BS=164) | 8-8 | 41.87 | 63.33 |

## C.6 Integration with Sparse Attention

The progressive quantization and the block-wise memory allocation can be applied naturally to sparse attention mechanisms. Calibration with positional interpolation is compatible with sparse attention mechanisms that rely on RoPE. We combine our method with QuestAttention (Tang et al., 2024) and evaluate the accuracy on DeepSeek-R1-Distill-Qwen-7B model. As shown in Table 10, our method does not degrade the pass@1 performance of QuestAttention.

Table 10: Results of PM-KVQ combined with QuestAttention. "BS" is short for "batch size".

| Models | Methods | AIME-2024 | |
|---|---|---|---|
| (Target GPU) | | pass@1 | Voting |
| DeepSeek-Qwen-7B (1×4090-24G) | - - | 41.04 | 63.33 |
| | QuestAttention | 33.33 | 63.33 |
| | QuestAttention+PM-KVQ | 33.58 | 60.00 |

## C.7 Efficiency Analysis of Pre-inference Process

Before the inference process, PM-KVQ performs block-wise memory allocation and calibration with positional interpolation as preparation. Following the experimental setup in Section 4.1, we measure the time required for these pre-inference procedures. As shown in Table 11, compared with QServe (Lin* et al., 2024), PM-KVQ leverages positional interpolation to reduce calibration

sequence length from 8,192 to 2,048 tokens, substantially reducing the calibration time by up to 77.21%. The additional block-wise memory allocation procedure account for 22.50–23.53% pre-inference time.

Table 11: Latency of block-wise memory allocation and calibration. "PI" is short for "Positional Interpolation".

| Model | Method | Calibration | | Memory Allocation | Time |
|---|---|---|---|---|---|
| | | w/o PI | w/ PI | | |
| DeepSeek-Qwen-7B | QServe (search $\alpha$) | ✓ | | | 52 min |
| | PM-KVQ (BS=40) | | ✓ | | 13 min |
| | PM-KVQ (BS=32) | | ✓ | ✓ | 17 min |
| DeepSeek-Qwen-32B | QServe (search $\alpha$) | ✓ | | | 187 min |
| | PM-KVQ (BS=16) | | ✓ | | 44 min |
| | PM-KVQ (BS=12) | | ✓ | ✓ | 57 min |
| DeepSeek-LLaMA-70B | QServe (search $\alpha$) | ✓ | | | 408 min |
| | PM-KVQ (BS=16) | | ✓ | | 93 min |
| | PM-KVQ (BS=12) | | ✓ | ✓ | 120 min |

## C.8 ANALYSIS OF PROGRESSIVE QUANTIZATION

Long-CoT tasks exhibit highly variable response lengths. For example, Table 12 summarizes the number of responses of DeepSeek-R1-Distill-LLaMA-8B falling into different length ranges.

Table 12: The number of responses falling into different length ranges.

| Model | Response Length | AIME-2024 | AIME-2025 |
|---|---|---|---|
| DeepSeek-LLaMA-8B | 0–4K | 40 | 63 |
| | 4–8K | 99 | 67 |
| | 8–16K | 138 | 144 |
| | 16–32K | 203 | 206 |

For shorter responses (e.g., <16K tokens in our experiments), our method keeps the KV cache at a higher bit-width than Fbit throughout decoding. This reduces cumulative quantization error compared with using Fbit KV cache, thereby preserving performance. For longer responses (e.g., >16K tokens in our experiments), although the KV cache eventually shrinks to Fbit, the first 16K tokens are generated using a higher-precision KV cache. Consequently, the model benefits from more accurate early-stage generations.

## D PROOF OF EQUIVALENT RIGHT SHIFT

**Theorem D.1** (Equivalent Right Shift). *Given a 16-bit floating-point tensor $\mathbf{X}_{BF16}$, let $\mathbf{X}_{2b}$ and $\mathbf{X}_b$ denote the 2b-bit and b-bit quantized tensors of $\mathbf{X}_{BF16}$, respectively. Then*

$$\mathbf{X}_b = \left( (2^{2b} - 2^b + 1)(\mathbf{X}_{2b} + 2^{b-1}) \right) >> 3b. \tag{13}$$

*Proof.* Let the zero points be $Z_{2b} = Z_b = Z$. According to Equation (2), the scaling factors are given by

$$S_{2b} = \frac{\max(\mathbf{X}_{BF16}) - Z}{2^{2b} - 1}, \quad S_b = \frac{\max(\mathbf{X}_{BF16}) - Z}{2^b - 1}. \tag{14}$$

It follows that

$$S_b = (2^b + 1)S_{2b}. \tag{15}$$

Define

$$\widetilde{\mathbf{X}}_{2b} = \frac{\mathbf{X}_{BF16} - Z}{S_{2b}}, \quad \widetilde{\mathbf{x}}_b = \frac{\mathbf{X}_{BF16} - Z}{S_b}. \tag{16}$$

Then the quantized tensors are obtained by rounding:

$$\mathbf{X}_{2b} = \left\lfloor \widetilde{\mathbf{X}}_{2b} \right\rceil, \quad \mathbf{X}_b = \left\lfloor \widetilde{\mathbf{X}}_b \right\rceil, \tag{17}$$

and we have

$$\widetilde{\mathbf{X}}_{2b} = (2^b + 1)\widetilde{\mathbf{X}}_b. \tag{18}$$

By the definition of rounding,

$$\mathbf{X}_{2b} - \frac{1}{2} \le \widetilde{\mathbf{X}}_{2b} < \mathbf{X}_{2b} + \frac{1}{2}. \tag{19}$$

Dividing both sides by $2^b + 1$ yields

$$\frac{\mathbf{X}_{2b} - \frac{1}{2}}{2^b + 1} \le \widetilde{\mathbf{X}}_b < \frac{\mathbf{X}_{2b} + \frac{1}{2}}{2^b + 1}. \tag{20}$$

Perform the Euclidean division of $\mathbf{X}_{2b}$ by $2^b + 1$:

$$\mathbf{X}_{2b} = q(2^b + 1) + r, \quad \text{with } 0 \le q \le 2^b - 1, 0 \le r \le 2^b. \tag{21}$$

Then,

$$q + \frac{r - \frac{1}{2}}{2^b + 1} \le \widetilde{\mathbf{X}}_b < q + \frac{r + \frac{1}{2}}{2^b + 1}. \tag{22}$$

Now consider the expression:

$$\begin{aligned}
\left((2^{2b} - 2^b + 1)(\mathbf{X}_{2b} + 2^{b-1})\right) >> 3b &= \left\lfloor \frac{(2^{2b} - 2^b + 1)(q(2^b + 1) + r + 2^{b-1})}{2^{3b}} \right\rfloor \\
&= q + \left\lfloor \frac{q + (2^{2b} - 2^b + 1)(r + 2^{b-1})}{2^{3b}} \right\rfloor.
\end{aligned} \tag{23}$$

We proceed by considering two cases for the remainder $r$:

**Case 1:** $0 \le r \le 2^{b-1}$.

Then,

$$q - \frac{1}{2} < q - \frac{\frac{1}{2}}{2^b + 1} \le \widetilde{\mathbf{X}}_b < q + \frac{2^{b-1} + \frac{1}{2}}{2^b + 1} = q + \frac{1}{2}. \tag{24}$$

Hence, rounding gives $\mathbf{X}_b = \left\lfloor \widetilde{\mathbf{X}}_b \right\rceil = q$.

Moreover,

$$\frac{q + (2^{2b} - 2^b + 1)(r + 2^{b-1})}{2^{3b}} \ge \frac{(2^{2b} - 2^b + 1) \cdot 2^{b-1}}{2^{3b}} > \frac{2^{2b} \cdot 2^{b-1}}{2^{3b}} = \frac{1}{2} > 0, \tag{25}$$

and

$$\begin{aligned}
\frac{q + (2^{2b} - 2^b + 1)(r + 2^{b-1})}{2^{3b}} &\le \frac{2^b - 1 + (2^{2b} - 2^b + 1)(2^{b-1} + 2^{b-1})}{2^{3b}} \\
&= 1 - \frac{(2^b - 1)^2}{2^{3b}} < 1.
\end{aligned} \tag{26}$$

Therefore,

$$\left\lfloor \frac{q + (2^{2b} - 2^b + 1)(r + 2^{b-1})}{2^{3b}} \right\rfloor = 0, \tag{27}$$

and thus

$$\left((2^{2b} - 2^b + 1)(\mathbf{X}_{2b} + 2^{b-1})\right) >> 3b = q = \mathbf{X}_b. \tag{28}$$

**Case 2:** $2^{b-1} + 1 \le r \le 2^b$.

Then,

$$q + \frac{1}{2} = q + \frac{2^{b-1} + 1 - \frac{1}{2}}{2^b + 1} \le \widetilde{\mathbf{X}}_b < q + \frac{2^b + \frac{1}{2}}{2^b + 1} < q + 1. \tag{29}$$

Thus, rounding gives $\mathbf{X}_b = \left\lfloor \widetilde{\mathbf{X}}_b \right\rceil = q + 1$.

Moreover,

$$\frac{q + (2^{2b} - 2^b + 1)(r + 2^{b-1})}{2^{3b}} \geq \frac{(2^{2b} - 2^b + 1)(2^{b-1} + 1 + 2^{b-1})}{2^{3b}} = \frac{2^{3b} + 1}{2^{3b}} > 1, \qquad (30)$$

and

$$\begin{aligned}
\frac{q + (2^{2b} - 2^b + 1)(r + 2^{b-1})}{2^{3b}} &\leq \frac{2^b - 1 + (2^{2b} - 2^b + 1)(2^b + 2^{b-1})}{2^{3b}} \\
&= \frac{2^{3b} + 2^{3b-1} - 2^{2b} - 2^{2b-1} + 2^{b+1} + 2^{b-1} - 1}{2^{3b}} \\
&= 2 - \frac{2^{3b-1} + 2^{2b} + 2^{2b-1} - 2^{b+1} - 2^{b-1} + 1}{2^{3b}} < 2.
\end{aligned} \qquad (31)$$

Therefore,

$$\left\lfloor \frac{q + (2^{2b} - 2^b + 1)(r + 2^{b-1})}{2^{3b}} \right\rfloor = 1, \qquad (32)$$

and thus

$$\left( (2^{2b} - 2^b + 1)(\mathbf{X}_{2b} + 2^{b-1}) \right) >> 3b = q + 1 = \mathbf{X}_b. \qquad (33)$$

In both cases, the desired equality holds, which completes the proof. $\qquad \square$

## E  LIMITATIONS

In this paper, we do not consider all of the attention mechanisms, such as the multi-head latent attention (MLA), which is quite different from the widely used Group-Query Attention (GQA). Besides, we do not combine the proposed PM-KVQ with other system-level optimization techniques and inference engines, which yields for future work.

