# OpenReview forum: "PM-KVQ: Progressive Mixed-precision KV Cache Quantization for Long-CoT LLMs"
_ICLR.cc/2026/Conference — ICLR 2026 Poster_

### Official Review · Reviewer_WmCs · 2025-10-26

**Soundness:** 2
**Presentation:** 3
**Contribution:** 2
**Rating:** 4
**Confidence:** 5

**Summary:**

This paper addresses KV cache quantization from a utility-driven perspective. The authors hypothesize that by fully utilizing the available memory and reducing the quantization bit-width only when the memory becomes full, one can achieve better CoT reasoning accuracy. Based on this idea, they propose a progressive quantization approach with an Equivalent Right Shift strategy. Specifically, all KV caches are stored in full precision initially, and once the memory limit is reached, the caches are progressively quantized by half using the proposed strategy. The authors further suggest assigning different memory budgets to each transformer block and formulate this allocation as an integer programming problem. To enable effective calibration for long contexts, they adopt positional interpolation to approximate long-context calibration using short-context data. Experimental results show that progressive quantization outperforms state-of-the-art fixed-precision quantization methods while maintaining comparable efficiency.

**Strengths:**

1. Approaching KV cache quantization from a utility-driven perspective is interesting and practically relevant, as it reflects real-world deployment considerations.

2. The figure illustrating the main idea of the paper is well-designed and easy to follow, effectively conveying the core concept.

3. The authors conduct experiments on multiple models, demonstrating the broad applicability and effectiveness of the proposed method.

**Weaknesses:**

1. My main concern lies in the fairness of the experimental results presented in Table 1. The baseline comparison is rather limited, as KIVI serves as the only comparable baseline in most evaluations. Since KIVI uses a fixed precision while the proposed method can employ higher precision during generation, the comparison may not be entirely fair. It would be helpful to report the memory usage during generation for both KIVI and the proposed method to provide a more complete picture. It remains unclear whether the observed improvement primarily comes from the fact that the proposed approach occupies more memory overall during generation, which actually contradicts the reason why we apply quantization, as we want to reduce the memory usage.

2. Because the proposed method depends on fully utilizing the available memory, it is important to evaluate its performance across different hardware configurations with varying memory capacities. Such an analysis would clarify how accuracy scales when the available memory changes.

3. I find the use of positional interpolation insufficient to address the main challenge of long-context calibration. This method essentially distributes a small number of tokens over a wide range, leaving many positions without properly calibrated data. Moreover, according to Table 4, positional interpolation appears to offer limited improvement, which further supports this concern.

**Questions:**

1. How does the method compare to KIVI that also fully utilizes the available memory? Would there be a way to conduct such a fair comparison?

2. For the Equivalent Right Shift strategy, is it reasonable to keep the zero point unchanged? Would it be possible for the zero point to vary across different quantization bit levels?

3. In Table 4, when s increases to 16, the use of positional interpolation appears to cause performance degradation. Does this imply that the proposed method has inherent limitations? Overall, the method’s effect on pass@1 seems marginal.

4. There is a typo around line 298: should it be CMIMC-2025 or CMIMC-2024?

---

> ### Author Response · Authors · 2025-11-23
>
> We sincerely appreciate reviewer WmCs’s positive feedback on our utility-driven perspective, the clarity of our main figure, and the extensive experiments demonstrating the method’s broad applicability and effectiveness. We address the concerns as follows.
>
> ## W1&Q1: Fair comparison with KIVI
> > My main concern lies in the fairness of the experimental results presented in Table 1. The baseline comparison is rather limited, as KIVI serves as the only comparable baseline in most evaluations. Since KIVI uses a fixed precision while the proposed method can employ higher precision during generation, the comparison may not be entirely fair. It would be helpful to report the memory usage during generation for both KIVI and the proposed method to provide a more complete picture. It remains unclear whether the observed improvement primarily comes from the fact that the proposed approach occupies more memory overall during generation, which actually contradicts the reason why we apply quantization, as we want to reduce the memory usage.
>
> > How does the method compare to KIVI that also fully utilizes the available memory? Would there be a way to conduct such a fair comparison?
>
> In our experiments, our method occupies more memory than KIVI before the KV cache bit-width is reduced to the final Fbit. However, under uniform Fbit settings, both our approach and KIVI consume the same amount of memory once the KV cache bit-width is reduced to Fbit.
>
> In fact, under the same GPU and batch size constraints, if KIVI adopts a higher quantization bit-width, it will consume more memory and therefore exhaust the available memory budget earlier, preventing the model from reaching the maximum output length. To ensure a fair comparison, we evaluate KIVI across different bit-width settings. When KIVI fully utilizes the memory budget, we truncate its output accordingly. As shown in the table below, our method still outperforms KIVI by 0.83%–18.33%.
>
> | Model            | Quantization Methods | Bit-width | AIME-2024 |        | AIME-2025 |        |
> | ---------------- | -------------------- | --------- | --------- | ------ | --------- | ------ |
> |                  |                      |           | pass@1    | Voting | pass@1    | Voting |
> | DeepSeek-Qwen-7B | --                   | 16-16     | 41.04     | 63.33  | 30.00     | 36.67  |
> |                  | KIVI (BS=32, 40)     | 16-16     | 21.88     | 36.67  | 21.67     | 26.67  |
> |                  | KIVI (BS=32, 40)     | 8-8       | 36.67     | 63.33  | 24.59     | 33.33  |
> |                  | KIVI (BS=32, 40)     | 4-4       | 39.38     | 63.33  | 26.46     | 36.67  |
> |                  | KIVI (BS=32, 40)     | 2-2       | 32.08     | 43.33  | 24.58     | 33.33  |
> |                  | PM-KVQ (BS=32)       | 4/2-4/2   | 40.21     | 66.67  | 28.96     | 40.00  |
> |                  | PM-KVQ (BS=40)       | 2-2       | 40.00     | 60.00  | 28.12     | 33.33  |
>
> ## W2: Performance across different hardware configurations
>
> > Because the proposed method depends on fully utilizing the available memory, it is important to evaluate its performance across different hardware configurations with varying memory capacities. Such an analysis would clarify how accuracy scales when the available memory changes.
>
> When the memory capacity of the target GPU increases, we can either maintain the per-request memory allocation by increasing the batch size or allocate a higher Fbit for each layer, allowing the KV cache to use higher bit-widths. We evaluate our method across different GPU memory capacities by increasing the Fbit accordingly. The results demonstrate that our method consistently achieves accuracy comparable to the original LLM under varying Fbit settings.
>
> | Model            | Target GPU | Quantization Methods | Bit-width | AIME-2024 |        |
> | ---------------- | ---------- | -------------------- | --------- | --------- | ------ |
> |                  |            |                      |           | pass@1    | Voting |
> | DeepSeek-Qwen-7B | --         | --                   | 16-16     | 41.04     | 63.33  |
> |                  | 1×4090-24G | PM-KVQ (BS=32)       | 4/2-4/2   | 40.21     | 66.67  |
> |                  |            | PM-KVQ (BS=40)       | 2-2       | 40.00     | 60.00  |
> |                  | 1×A100-40G | PM-KVQ (BS=36)       | 8/4-8/4   | 40.63     | 63.33  |
> |                  |            | PM-KVQ (BS=58)       | 4-4       | 40.83     | 66.67  |
> |                  | 1×A100-80G | PM-KVQ (BS=108)      | 16/8-16/8 | 42.50     | 66.67  |
> |                  |            | PM-KVQ (BS=164)      | 8-8       | 41.87     | 63.33  |

---

> ### Author Response · Authors · 2025-11-23
>
> ## W3: Effect of positional interpolation
> > I find the use of positional interpolation insufficient to address the main challenge of long-context calibration. This method essentially distributes a small number of tokens over a wide range, leaving many positions without properly calibrated data. Moreover, according to Table 4, positional interpolation appears to offer limited improvement, which further supports this concern.
>
> The purpose of calibration is to capture the data distribution. As shown in Figure3(c) middle, when using long-context calibration data, the calibration results show that the data in this channel of Key Cache shows sine curve and reaches peak value as the context gets longer. As shown in Figure 3(c) bottom, these features can also be obtained by using short-context calibration data, although some positions are not calibrated.
>
> The effect of positional interpolation mainly focuses on those long responses. For example, in Table 4, calibration with positional interpolation brings 1.66% improvement on the whole dataset. However, on responses with context length of 16K~32K, positional interpolation improves pass@1 by 8.50%.
>
> | Model             | Calibration Sequence Length | Position Scaling  Factor | Effective Length | 0~4K pass@1 | 16K~32K pass@1 |
> | ----------------- | --------------------------- | ------------------------ | ---------------- | ----------- | -------------- |
> | DeepSeek-LLaMA-8B | 2048                        | 1                        | 2048             | 100.00      | 4.54           |
> |                   | 2048                        | 4                        | 8192             | 100.00      | 13.04          |
>
>
>
> ## Q2: Zero-point handling under Equivalent Right Shift
>
> > For the Equivalent Right Shift strategy, is it reasonable to keep the zero point unchanged? Would it be possible for the zero point to vary across different quantization bit levels?
>
> For the Equivalent Right Shift strategy, our goal is to ensure equivalence between the following two quantization procedures:  (a) Directly quantize a BF16 tensor to b-bit. (b) First quantize a BF16 tensor to 2b-bit and then shrink the bit-width to b-bit using Equivalent Right Shift strategy. In procedure (a), the zero point is defined as $Z=\text{min}(X_{\text{BF16}})$. In procedure (b), when the BF16 tensor $X_{\text{BF16}}$ is quantized to 2b-bit, the zero point $Z=\text{min}(X_{\text{BF16}})$. So during the Equivalent Right Shift strategy, the zero point is kept unchanged. The zero point is shared in a quantization group containing different quantization levels, so it cannot vary across different quantization bit levels.
>
> ## Q3: Performance drop when s=16
>
> > In Table 4, when s increases to 16, the use of positional interpolation appears to cause performance degradation. Does this imply that the proposed method has inherent limitations? Overall, the method’s effect on pass@1 seems marginal.
>
> Thanks for pointing out the performance drop when *s* increases to 16. We agree that this clearly indicates that the computational savings of positional interpolation are not unlimited, and overly aggressive scaling can indeed degrade performance. We have added clarifications in the revised version of the paper.
>
> At the same time, within a moderate range (e.g. s=4), the method consistently preserves model performance. This means that the calibration data length can be reduced to one quarter of the original while maintaining comparable pass@1 accuracy. In this regime, positional interpolation effectively reduces computational cost without sacrificing quality.
>
> ## Q4: Typo of CMIMC
>
> > There is a typo around line 298: should it be CMIMC-2025 or CMIMC-2024?
>
> Thanks for pointing out the mistake. We conduct our experiments on CMIMC-2024 dataset and we have fixed this typo.

---

### Official Review · Reviewer_u27E · 2025-10-30

**Soundness:** 3
**Presentation:** 2
**Contribution:** 2
**Rating:** 4
**Confidence:** 4

**Summary:**

The paper proposes PM-KVQ, a post-training quantization scheme for long-CoT LLM inference that combines: a. progressive quantization (start at higher precision and shrink the KV cache bit-width only when memory is about to run out), including an “Equivalent Right Shift” rule (`Eq. 3`) for precise bit-width shrinking, b. block-wise memory allocation, cast as a small integer program solved with CVXPY to allocate higher bit-widths to more sensitive transformer blocks; and c. calibration with positional interpolation to expose short calibration sequences to long-ctx RoPE phases. Reported results on DeepSeek-R1-Distill (7B-70B) and QwQ-32B show improved pass@1/vote accuracy over KIVI/RotateKV/MiKV at 2-4-bit KV cache, with throughput close to KIVI but below it.

**Strengths:**

- Clear diagnosis of two long-CoT pain points (cumulative error, RoPE low-frequency channels), with concrete formulations (Eqs. 9-12) motivating the positional interpolation trick.
- Simple but effective shrinking rule (Eq. 3) that avoids round-trip dequantization in implementation.
- Block-wise allocation objective is standard, implementable, and explains gains when memory is partially free.

**Weaknesses:**

- The paper only reports FP16 results, omitting bf16, which is the de facto standard for inference. Since bf16 offers wider dynamic range and distinct hardware behavior, excluding it leaves uncertainty about PM-KVQ’s performance and compatibility in realistic deployment settings.
- Accuracy under fake quant: Reporting accuracy without real 2-bit/4-bit kernels weakens the claim that PM-KVQ is robust in practice.

**Questions:**

- **L41**: What is the reference for the claim “to generate 128K tokens”?
- **Table 1**: I am skeptical of using **BS** in your setting. What is the number of output tokens? Since the reasoning model generates long-CoT with the mentioned BS and GPU memory, it’s most likely that we get CUDA OOM.
- **Table 1**: **QwQ** has 32B parameters. How do you use it with one A100?
- **Sec 3.2**: Why is **CVXPY** used? What was the reason behind that?
- **Sec 4.1**: Your evaluation metric is **pass@k** where *k = 1*. What is *n* (number of independent trials)?
- **Sec 4.1**: Regarding **Voting**, how many samples did you draw?
- Could the progressive quantization step, while helpful for memory control, also hinder GPU utilization by blocking parallel execution?

---

> ### Author Response · Authors · 2025-11-23
>
> We sincerely thank reviewer u27E for recognizing our motivation behind our positional-interpolation formulation, the practicality of our shrinking rule, and the effectiveness of our block-wise allocation strategy. We address the concerns as follows.
>
> ## W1: BF16 evaluation
>
> > The paper only reports FP16 results, omitting bf16, which is the de facto standard for inference. Since bf16 offers wider dynamic range and distinct hardware behavior, excluding it leaves uncertainty about PM-KVQ’s performance and compatibility in realistic deployment settings.
>
> Sorry for the oversight in our writing. All of our experiments were in fact conducted using the BF16 format, which we have now correctly reflected in the revised version of the paper.
>
> ## W2: Accuracy under real quantization
>
> > Accuracy under fake quant: Reporting accuracy without real 2-bit/4-bit kernels weakens the claim that PM-KVQ is robust in practice.
>
> We evaluate accuracy using actual 16/8/4/2-bit CUDA kernels rather than relying solely on fake quantization. The results show that real low-precision kernels yield accuracy comparable to fake quantization.
>
> | Models            | Quantization Methods | AIME-2024 |        | AIME-2025 |        |
> | ----------------- | -------------------- | --------- | ------ | --------- | ------ |
> |                   |                      | pass@1    | Voting | pass@1    | Voting |
> | DeepSeek-Qwen-7B | PM-KVQ-fake (BS=40)  | 40.00     | 60.00  | 28.12     | 33.33  |
> |                   | PM-KVQ-real (BS=40)  | 39.79     | 60.00  | 28.75     | 33.33  |

---

> ### Author Response · Authors · 2025-11-23
>
> ## Q1: Lack of reference
>
> > L41: What is the reference for the claim “to generate 128K tokens”?
>
> The DeepSeek-R1-Distill series [1] models have a maximum context length of 128K tokens. We have added the appropriate reference to the revised version of the paper.
>
> ## Q2&Q3: GPU memory usage
>
> > Table 1: I am skeptical of using BS in your setting. What is the number of output tokens? Since the reasoning model generates long-CoT with the mentioned BS and GPU memory, it’s most likely that we get CUDA OOM.
>
> > Table 1: QwQ has 32B parameters. How do you use it with one A100?
>
> Taking the QwQ-32B model as an example. The weights of QwQ-32B occupy $$(32\times 2^{30}) \ \text{parameters}\times 2 \ \text{byte/parameter}\times 2^{-30} \ \text{byte/GB}=64 \ \text{GB}$$ of GPU memory. With batch size 16, context length 32K, and 2-bit KV-cache quantization, the KV-cache memory cost is:$$16 \ \text{requests}\times (32\times2^{10}) \ \text{tokens/(request}\cdot\text{layer)}\times64 \ \text{layers}\times2048 \ \text{numbers/token}\times2 \ \text{bit/number}\times\frac{1}{8\times2^{30}} \ \text{GB/bit}=16 \ \text{GB}.$$The overall memory consumption is $$64 \  \text{GB}+16  \ \text{GB}=80 \  \text{GB}, $$ which can fit within a single A100-80G GPU.
>
> ## Q4: Selection of optimization tools
>
> > Sec 3.2: Why is CVXPY used? What was the reason behind that?
>
> CVXPY is an open-source framework designed for formulating and solving convex optimization problems, and it has also been adopted by other mixed-precision quantization methods such as LLM-MQ [2]. We follow this established practice for consistency and ease of experimentation. In addition, we verified our results using the Pyomo [3] framework with the GLPK [4] solver and obtained the same memory-allocation solutions.
>
> ## Q5&Q6: Number of samples
>
> > Sec 4.1: Your evaluation metric is pass@k where k = 1. What is n (number of independent trials)?
>
> > Sec 4.1: Regarding Voting, how many samples did you draw?
>
> When calculating pass@1 and voting accuracy, we generate 16 independent samples for each mathematical problem and 4 independent samples for each code generation problem.
>
> ## Q7: Potential GPU utilization degradation
>
> > Could the progressive quantization step, while helpful for memory control, also hinder GPU utilization by blocking parallel execution?
>
> In our experiments in Section 4.3, all requests within a batch arrive simultaneously and share the same number of input tokens. As a result, the bit-width shrinking step for each request occurs at the same decoding step, ensuring that the process is synchronized across the batch. Once the shrinking step is completed, all requests continue to run in parallel without reducing GPU utilization.
>
> In practical scenarios where requests may arrive asynchronously, a request that requires bit-width shrinking might temporarily leave the parallel execution group, perform the shrinking step, and then rejoin parallel execution once the process is complete. Achieving optimal GPU utilization in such settings may require integration with system-level optimizations.
>
>
>
> [1] Guo, Daya, et al. "Deepseek-r1: Incentivizing reasoning capability in llms via reinforcement learning." *arXiv preprint arXiv:2501.12948* (2025).
>
> [2] Li, Shiyao, et al. "Llm-mq: Mixed-precision quantization for efficient llm deployment." *The Efficient Natural Language and Speech Processing Workshop with NeurIPS*. Vol. 9. 2023.
>
> [3] Hart, William E., Jean-Paul Watson, and David L. Woodruff. "Pyomo: modeling and solving mathematical programs in Python." *Mathematical Programming Computation* 3.3 (2011): 219-260.
>
> [4] GNU Linear Programming Kit. http://www.gnu.org/software/glpk/glpk.html

---

### Official Review · Reviewer_nh48 · 2025-10-31

**Soundness:** 3
**Presentation:** 3
**Contribution:** 3
**Rating:** 6
**Confidence:** 5

**Summary:**

This paper propose progressive mixed precision KV Cache quantization, of which a higher bit-width is assigned to more sensitive transformer blocks tailored for long-CoT scenarios, aiming for the low memory usage and quantization error. Extensive experiments on 7B–70B long-CoT LLMs show that the proposed block-wise and mixed precision quantization method improves reasoning benchmark performance by up to 8% over SOTA baselines under the same memory budget and achieves 2.73–5.18× throughput over the original 16-bit LLMs.

**Strengths:**

The progressive mixed precision quantization is interesting: 1) initially, the high bit (16-bit) quantization is used for the short-sequence; 2) then progressively shrink the bit width, while the high sensitive transformer blocks are maintained with the high bit width to narrow the quantization error; 2) and the memory allocation is block-wise, which is adaptive to the PageAttention.

Secondary, the experiments show good, especially for the long-cot tasks, the proposed method improves reasoning benchmark performance by up to 8% over SOTA baselines.

**Weaknesses:**

The mixed precision quantization of KV cache is mature in the academia, such as KVTuner which allocate different bit width for different layer and K/V by optimized search algorithm. So the comparison with SOTA mixed quantization methods is not enough. And the practical benefit on the hardware is not given, such as the memory access saving and the throughput increase.

**Questions:**

1. How can this method used with PageAttention, for the quantized KV Cache management?
2. How can this method used with sparse Attention, such QuestAttention, DSA or NSA?
3. Can you draw the theoretical analysis, why such progressive shrinking is useful for long-cot tasks?

---

> ### Author Response · Authors · 2025-11-23
>
> We sincerely appreciate the reviewer nh48’s positive evaluation of our progressive mixed-precision quantization design and  experimental results. We address the concerns as follows.
>
> ## W1: Comparison with stronger baseline
>
> > The mixed precision quantization of KV cache is mature in the academia, such as KVTuner which allocate different bit width for different layer and K/V by optimized search algorithm.
>
> We conduct additional experiments comparing our method with KVTuner [1] on DeepSeek-R1-Distill-Qwen-7B under the same target GPU and batch size settings. For KVTuner, we adopt per-channel asymmetric quantization for the Key cache and per-token asymmetric quantization for the Value cache. Our results show that the pass@1 of our method surpasses KVTuner by 2.71-6.04%.
>
> | Model            | Quantization Methods | Bit-width   | AIME-2024 |        | AIME-2025 |        |
> | ---------------- | -------------------- | ----------- | --------- | ------ | --------- | ------ |
> |                  |                      |             | pass@1    | Voting | pass@1    | Voting |
> | DeepSeek-Qwen-7B | --                   | 16-16       | 41.04     | 63.33  | 30.00     | 36.67  |
> |                  | KIVI (BS=32)         | 2-2         | 32.08     | 43.33  | 24.58     | 33.33  |
> |                  | KVTuner (BS=32)      | 8/4/2-8/4/2 | 34.17     | 56.67  | 26.25     | 33.33  |
> |                  | PM-KVQ (BS=32)       | 4/2-4/2     | 40.21     | 66.67  | 28.96     | 40.00  |
>
> ## W2: Lack of hardware benefit
>
> > And the practical benefit on the hardware is not given, such as the memory access saving and the throughput increase.
>
> We estimate the reduction in memory access by computing the average bit-width of the accessed KV cache. For simplicity, we assume each layer’s Fbit is 2, the input length is 1 token, and the output length is 32,768 tokens. Under these settings, the average bit-width of the accessed KV cache is $$\frac{1}{\sum_{i=1}^{32768}i}\left(\sum_{i=1}^{4096}16i+\sum_{i=4097}^{8192}8i+\sum_{i=8193}^{16384}4i+\sum_{i=16385}^{32768}2i\right)=2.875$$ per request. Compared to the original BF16 KV cache, this corresponds to an 82% reduction in memory access. We also report practical throughput in Table 2 of our paper. Our method achieves a 2.73-5.18× throughput improvement over the original 16-bit LLMs.
>
>
> [1] Li, Xing, et al. "KVTuner: Sensitivity-Aware Layer-Wise Mixed-Precision KV Cache Quantization for Efficient and Nearly Lossless LLM Inference." *Forty-second International Conference on Machine Learning*.

---

> ### Author Response · Authors · 2025-11-23
>
> ## Q1: Integration with PagedAttention
>
> > How can this method used with PageAttention, for the quantized KV Cache management?
>
> Our method is compatible with PagedAttention [1] through a straightforward adaptation of page management. Specifically, we can fix the memory size of each page, allowing the number of tokens per page to vary with the quantization bit-width. For example, a page that can store N 16-bit tokens can alternatively store 2N 8-bit tokens, 4N 4-bit tokens, or 8N 2-bit tokens. During the bit-width shrinking process, this design naturally supports dynamic page consolidation: when the bit-width is reduced, every two adjacent pages can be merged into a single page. This ensures efficient KV cache organization.
>
> ## Q2: Integration with sparse attention
>
> > How can this method used with sparse Attention, such QuestAttention, DSA or NSA?
>
> The **progressive quantization** and the **block-wise memory allocation** can be applied naturally to sparse attention mechanisms such as QuestAttention [2], DSA [3], and NSA [4]. In all these methods, the KV cache expands as the generated sequence length increases. Consequently, we can first profile the sensitivity of each block prior to inference and allocate an appropriate Fbit to each block. During inference, we progressively reduce the KV-cache precision from 16-bit down to the allocated Fbit, enabling seamless integration with sparse attention.
>
> **Calibration with positional interpolation** is compatible with sparse attention mechanisms that rely on RoPE, such as QuestAttention and DSA. However, the positional encoding of NSA is not publicly disclosed, so combining positional interpolation with NSA may not yield correct behavior.
>
> We combine our method with QuestAttention and evaluate the accuracy on DeepSeek-R1-Distill-Qwen-7B model. As shown in the following table, our method does not degrade the pass@1 performance of QuestAttention.
>
> | Model            | Methods                         | AIME-2024 |        |
> | ---------------- | ------------------------------- | --------- | ------ |
> |                  |                                 | pass@1    | Voting |
> | DeepSeek-Qwen-7B | --                              | 41.04     | 63.33  |
> |                  | QuestAttention                  | 33.33     | 63.33  |
> |                  | QuestAttention + PM-KVQ (BS=32) | 33.58     | 60.00  |
>
> ## Q3: Theoretical analysis of progressive quantization
>
> > Can you draw the theoretical analysis, why such progressive shrinking is useful for long-cot tasks?
>
> Long-CoT tasks exhibit highly variable response lengths. For example, the following table summarizes the number of responses of DeepSeek-R1-Distill-LLaMA-8B falling into different length ranges.
>
> | Response length | AIME-2024 | AIME-2025 |
> | --------------- | --------- | --------- |
> | 0~4K            | 40        | 63        |
> | 4K~8K           | 99        | 67        |
> | 8K~16K          | 138       | 144       |
> | 16K~32Ks        | 203       | 206       |
>
> For shorter responses (e.g., <16K tokens in our experiments), our method keeps the KV cache at a higher bit-width than Fbit throughout decoding. This reduces cumulative quantization error compared with using Fbit KV cache, thereby preserving performance. For longer responses (e.g., >16K tokens in our experiments), although the KV cache eventually shrinks to Fbit, the first 16K tokens are generated using a higher-precision KV cache. Consequently, the model benefits from more accurate early-stage generations.
>
>
> [1] Kwon, Woosuk, et al. "Efficient memory management for large language model serving with pagedattention." *Proceedings of the 29th symposium on operating systems principles*. 2023.
>
> [2] Tang, Jiaming, et al. "QUEST: query-aware sparsity for efficient long-context LLM inference." *Proceedings of the 41st International Conference on Machine Learning*. 2024.
>
> [3] DeepSeek-AI. "DeepSeek-V3.2-Exp: Boosting Long-Context Efficiency with DeepSeek Sparse Attention."
>
> [4] Yuan, Jingyang, et al. "Native sparse attention: Hardware-aligned and natively trainable sparse attention." *Proceedings of the 63rd Annual Meeting of the Association for Computational Linguistics (Volume 1: Long Papers)*. 2025.

---

> > ### Comment · Reviewer_nh48 · 2025-11-24
> >
> > Thank you for your reply

---

> > > ### Author Response · Authors · 2025-11-24
> > > **Thanks for the feedback**
> > >
> > > Thank you very much for your thoughtful comments. We are happy to discuss any aspect of the paper further at any time.

---

### Meta-Review · Area_Chair_Bmys · 2026-01-05

**Summary:**

While the reviewers acknowledged the novelty of the technical design and the performance of the proposed quantization approach, they also asked for more comprehensive experimental comparisons with the baselines and in more practical settings. In particular -

Reviewer nh48 asked for a comparison with KVTuner and a practical benefit analysis, such as memory and throughput. They also asked for a theoretical analysis providing the rationales behind the benefit of progressive shrinking in long CoT tasks.

Reviewer u27E expressed concerns over some evaluation settings, including FP16 instead of bf16 and the fake quant results. They also asked many implementation detail questions such as A100 memory usage, number of samples for voting, and potential GPU utilization degradation issues.

Reviewer WmCs pointed out that the comparison with the baseline can be potentially unfair, because the proposed method uses more memory. Accordingly, they asked for experiments over a wider range of hardware settings. They also expressed doubts over the effectiveness of positional interpolation.

**Reviewer Concerns:**

Most reviewer concerns are addressed. In particular -

**Reviewer nh48:** Their concern over stronger baselines is well addressed with the new experiments KV tuner.  The authors also provided a theoretical analysis of memory usage reduction and pointed out that Table 2 already provides the analysis of practical throughput improvement, although an empirical analysis on the memory usage is preferable. The authors provided a high-level explanation of why progressive shrinking is beneficial even in long CoT tasks, although the reviewer might be looking for a more formal analysis.

**Reviewer u27E:** It turns out that the FP16 setting was a mistake in writing. The concern about fake quant results is well-addressed with new results using real quant. Implementation detail questions are well answered by the reviewer.

**Reviewer WmCs:** The authors provide abundant new results showing the comparison with the baseline under different constraint settings, as well as new results under different hardware settings. The authors clarified that positional interpolation shows most benefit in long CoT generations.

**Reviewer Scores:**

Reviewer nh48 may maintain their score (6) because the response does not fully address the concerns.

Reviewers 627E and WmCs may increase their score to 6 because of the abundant new experiment results.

After the potential score change, this paper becomes a borderline paper leaning towards acceptance. I would recommend acceptance of the paper due to its solid technical novelty and strong performance compared to baselines.

---

### Decision · Program_Chairs · 2026-01-26

Accept (Poster)